# A Differentiable Sequence Model Perspective on Policy Gradients

## Abstract

Progress in sequence modeling with deep learning has been driven by the advances in temporal credit assignment coming from better gradient propagation in neural network architectures. In this paper, we reveal that using deep dynamics models conditioned on sequences of actions allows to draw a direct connection between gradient propagation in neural networks and policy gradients, and to harness those advances for sequential decision-making. We leverage this connection to analyze, understand and improve policy gradient methods with tools that have been developed for deep sequence models, theoretically showing that modern architectures provably give better policy gradients. Furthermore, we empirically demonstrate that, in our algorithmic framework, better sequence models entail better policy optimization: when the environment dynamics is well-behaved, we find that better neural network architectures yield more accurate policy gradients; when it is chaotic or non-differentiable, we discover that neural networks are able to provide gradients better-suited for policy optimization compared to the real differentiable simulator. On an optimal control testbed, we show that, within our framework, agents enjoy increased long-term credit assignment capabilities and sample efficiency when compared to traditional model-based and model-free approaches.

## 1 Introduction

In reinforcement learning (RL), an agent executes actions in an environment, receiving a sequence of rewards with the goal of maximizing their sum, the return. On a mechanistic level, this problem resembles the one of sequence modeling: after processing each input, the sequence model incurs a loss for a prediction, and its overall goal is to minimize the cumulative loss obtained on a sequence. Fundamentally, both problems are concerned with the challenge of temporal credit assignment, wherein credit must be assigned to the appropriate inputs responsible for a given reward or loss. In either case, credit can be assigned by estimating the gradient of the performance with respect to the inputs of interest. Acknowledging this parallel is not just an exercise in correspondence but, as we show in this paper, a powerful approach to deriving efficient RL algorithms.

In sequence modeling, backpropagation through time has remained the default method for computing the gradient of the loss function for over thirty years (Werbos, 1988). Progress has been mainly driven by the development of neural network architectures exhibiting better gradient propagation properties (Hochreiter & Schmidhuber, 1997; Cho et al., 2014), leading to the successful modeling of long-term dependencies thousands of time steps apart (Vaswani et al., 2017; Gu et al., 2022).

If we follow the parallel from sequence modelling to RL, a natural technique to learn a policy is to treat the computational graph created by the interaction between the agent and the dynamics of the environment in a similar way, and to compute a *policy gradient* by differentiating through it. Such an approach has been a foundational aspect of classic RL methods (Werbos, 1974; Barto et al., 1983; Miller et al., 1995), and a component of successful deep RL algorithms (Hafner et al., 2022). However, backpropagation in this manner is seldom employed beyond a few dozen steps of the dynamics due to unstable or inaccurate gradients (Heess et al., 2015). Rather, deep RL methods have mostly sidestepped the problem by relying on temporal difference learning, which is commonly restricted to an effective credit assignment horizon that rarely exceeds a hundred time steps (Ni et al., 2023).

Despite the success of sequence modeling and its similarities with policy gradients, there is still limited understanding of neural networks' role in estimating policy gradients. Can decades of scien-

tific advances regarding gradient propagation in deep sequence models guide our understanding of policy gradient estimation via backpropagation? Could neural network architectures developed for sequence modeling lead to better policy gradients?

This paper aims to offer a positive answer to these questions. We do so by developing a framework based on the idea of computing policy gradients by differentiating through an *action-sequence model*: a predictor of the state dynamics of an environment conditioned on a sequence of actions without environment states. This formulation allows us to draw a direct connection between gradient propagation properties for a particular neural network architecture and the quality of the resulting policy gradient. We show that common failure cases encountered in policy gradient methods, such as the ones coming from estimating gradients through learned Markovian models (Heess et al., 2015), can be understood through the lens of results for gradient propagation in sequence models.

We argue that using action-sequence models for policy optimization has inherent advantages. Unlike traditional models conditioned on the entire history of states and actions, action-sequence models directly enjoy the gradient propagation properties of their core neural network architecture, leading to policy gradients more amenable to policy optimization. In this work, we demonstrate these properties both theoretically and empirically.

Theoretically, we show that modern sequence models, such as transformers (Vaswani et al., 2017), provably lead to better policy gradients when employed as a backbone for an action-sequence model. By interpreting model-based policy gradients differentiated through traditional dynamics models (e.g., Markovian) under the light of action-sequence models, we ground explanations of their conceptual and theoretical limitations on the existing theory of neural networks.

We complement the theoretical findings with in-depth empirical investigations. We show that using appropriate neural network architectures for action-sequence models yields policy gradients that are accurate in the presence of a well-behaved environment. Furthermore, we show that these models *can be better than the ones coming from the actual simulator* in the case of chaotic or non-differentiable dynamics. This provides an answer to open questions from recent work, which conjectured that using better neural network models might be a remedy for the challenge of policy optimization via backpropagation in the presence of non-smooth differentiable simulators (Metz et al., 2021; Suh et al., 2022). We demonstrate on a suite of realistic optimal control tasks (Howe et al., 2022) that the enhanced gradient propagation properties coming from appropriate neural network architectures used as action-sequence models allow for more effective policy optimization. In particular, we find they unlock longer temporal credit assignment capabilities, leading to an increased ability to solve long-horizon tasks and improved sample efficiency compared to traditional model-based and model-free approaches.

## 2 BACKGROUND

**Sequence modeling** Sequence modeling is a supervised learning problem concerned with predicting the sequence of labels $y_1, \ldots, y_T$ given an input sequence $x_0, \ldots, x_{T-1}$. A sequence model is any parameterized function function $g : \mathcal{X}^t \to \mathcal{Y}$ such that $g(x_0, \ldots, x_{t-1}) \approx y_t$. This formalism has been adopted in many applications, such as speech recognition, text generation and sentiment analysis. When $g$ is parameterized as a neural network, a similar strategy is used for training the model regardless of its specific architecture. We will mainly consider three types of sequence models: simple Recurrent Neural Networks (RNNs) (Werbos, 1974), Long Short-Term Memory networks (LSTMs) (Hochreiter & Schmidhuber, 1997) and attention-based models (e.g., transformers (Vaswani et al., 2017)).

**Problem definition** We are interested in deterministic discrete-time finite-horizon Markov Decision Processes (MDPs) (Fairbank, 2014), defined as $\mathcal{M} = (\mathcal{S}, \mathcal{A}, f, r, H, s_1)$, where $\mathcal{S} \subseteq \mathbb{R}^n$ is the state space, $\mathcal{A} \subseteq \mathbb{R}^m$ is the action space, $f : \mathcal{S} \times \mathcal{A} \to \mathcal{S}$ is the differentiable transition dynamics, $r : \mathcal{S} \to \mathbb{R}$ is the differentiable reward function, $H$ is the horizon and $s_1 \in \mathcal{S}$ is the initial state. The behavior of an agent in the environment is described by a policy $\pi_{\boldsymbol{\theta}} : \mathcal{S} \to \mathcal{A}$, belonging to a space of parameterized differentiable deterministic[1] stationary Markov policies $\Pi = \{\pi_{\boldsymbol{\theta}} : \boldsymbol{\theta} \in \mathbb{R}^d\}$.

---

[1]This assumption can be relaxed to a stochastic policy, where its derivative can be estimated using the reparameterizations (Kingma & Welling, 2014).

Without loss of generality, we omit possible time dependencies in the reward function and dynamics, but this has no impact on any of the results that follow.

**Gradient-based policy optimization** The goal of the agent is to maximize the cumulative reward of trajectories induced by its policy. In the deterministic setting, it corresponds to the maximization of the objective $J(\boldsymbol{\theta}; H) := \sum_{t=1}^{H} r(s_t)$, where $s_t = f(s_{t-1}, a_{t-1})$, $a_t = \pi_{\boldsymbol{\theta}}(s_t)$ and $s_1$ is given by the MDP. Policy gradient methods learn a policy by gradient descent using an estimation of $\nabla_{\boldsymbol{\theta}} J$. In this paper, we are interested in methods that compute the policy gradient by directly differentiating through the reward function, the transition dynamics, and a policy under non-restrictive smoothness assumptions. We consider the *open-loop policy gradient* defined below for the remainder of this work.

**Definition 1.** (Open-loop Policy Gradient under a deterministic MDP). *Let $r$ and $f$ be the differentiable reward and transition functions of a Markov Decision Process $\mathcal{M}$, $\Pi_{\boldsymbol{\theta}}$ a parametric space of differentiable deterministic policies and $s_t = f(s_{t-1}, a_{t-1})$. Given $\pi_{\boldsymbol{\theta}} \in \Pi_{\boldsymbol{\theta}}$, the* Open-loop Policy Gradient *of $\pi_{\boldsymbol{\theta}}$ under $\mathcal{M}$ is defined as:*

$$\nabla_{\boldsymbol{\theta}} J(\boldsymbol{\theta}; H) = \sum_{t=1}^{H} \frac{\partial r(s_t)}{\partial s_t} \sum_{k=1}^{t-1} \frac{\partial f(s_k, a_k)}{\partial a_k} \frac{\partial \pi_{\boldsymbol{\theta}}(s_k)}{\partial \boldsymbol{\theta}} \left( \prod_{i=k+1}^{t-1} \frac{\partial f(s_i, a_i)}{\partial s_i} \right).$$

Rooted in an open-loop policy, where actions depend only on $t$, this definition retains the benefits of a closed-loop policy during inference while simplifying gradient analysis and calculation. Practically, this gradient is often used in gradient-based policy optimization in model-based methods (Hafner et al., 2022; 2023) and can be evaluated by treating the policy inputs as a constant. For a more complete discussion of this gradient and its relationship with a standard policy gradient, see Appendix B.

When the dynamics $f$ are not known, the policy can be updated using a learned transition function $\hat{f}_{\boldsymbol{\psi}}(s_t, a_t)$ belonging to a space of parameterized differentiable functions $\mathcal{F} = \{f_{\boldsymbol{\psi}} : \boldsymbol{\psi} \in \mathbb{R}^{d_{\psi}}\}$. Given transitions $s_t, a_t, s_{t+1}$ sampled from the environment, the approximate one-step model $\hat{f}$, or Markovian model, can be learned with the following mean squared error loss

$$\mathcal{L}(\boldsymbol{\psi}; s_t, a_t, s_{t+1}) = (s_{t+1} - \hat{f}_{\boldsymbol{\psi}}(s_t, a_t))^2 . \tag{1}$$

This model can then be learned by gradient descent and used to compute the policy gradient.

## 3 A SEQUENCE MODELING PERSPECTIVE ON POLICY GRADIENTS

In this section, we present a differentiable sequence modeling perspective on policy gradients. The core of this proposal revolves around dynamics models conditioned on sequences of actions, which can be instantiated with any neural network architecture. When using such models, the long-term properties of the policy gradient are a direct result of the chosen architecture. In the rest of the paper, we will require the following assumption.

**Assumption 1.** *The reward function $r$ is $L_r$-Lipschitz and any policy $\pi_{\boldsymbol{\theta}} \in \Pi$ is $L_{\pi}$-Lipschitz.*

We use this standard assumption, commonly employed when backpropagating through the environment dynamics and rewards (Clavera et al., 2020), to prove our results on gradient propagation.

### 3.1 ACTION-SEQUENCE MODELS

Let us reflect on the mechanism linking a sequence of actions to its execution in the environment. If we think of the environment as a black-box machine, what the agent observes is that when presented with a sequence of actions $a_1, \ldots, a_{t-1}$, this machine produces a sequence of states $s_2, \ldots, s_t$. Under this perspective, the correspondence with sequence modeling is apparent. In that setting, without any knowledge about the mechanism linking a sequence of inputs $x_0, \ldots, x_{T-1}$ to a sequence of outputs $y_1, \ldots, y_T$, we want to learn a model $g$ able to predict the latter from the former; in an identical manner, an agent that wants to successfully interact with an environment, with no prior knowledge about its dynamics, needs to learn which states it will observe after executing a sequence of actions.

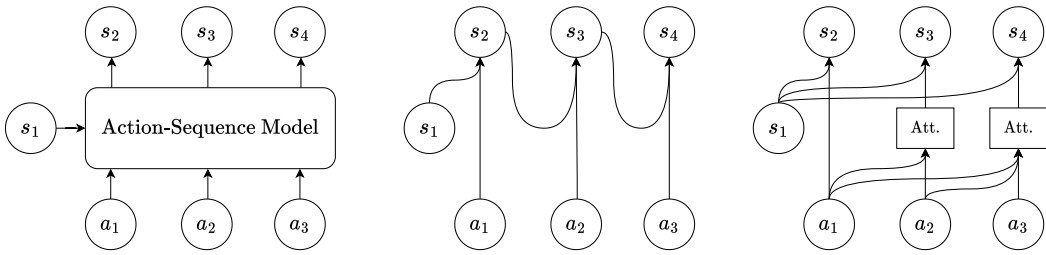

Figure 1: (**Left**): a generic representation of an ASM. (**Center**): an ASM instantiated with an RNN, which recovers the familiar computational graph of an MDP. (**Right**): an ASM instantiated with an attention-based model with attention modules (Att.). Gradient propagation from states back to actions will directly follow from the inherent properties of the underlying neural network architecture.

To reuse advances in sequence modeling for understanding and improving sequential decision-making agents, we want to build a model that follows the natural parallel we highlighted. This is a model of the type $g : \mathcal{A}^t \to \mathcal{S}; a_{1:t} \mapsto g(a_{1:t})$ that, given a sequence of actions, predicts each state resulting from executing those actions: we call this an *action-sequence model* (ASM). Without loss of generality, the dependency of the model from $s_1$ can be ignored in an MDP with a fixed initial state. Given an ASM, a policy can be learned by directly differentiating through it to compute the following policy gradient.

**Definition 2** (Open-loop Policy Gradient under an action-sequence model). *Let $g$ be an action-sequence model, $r$ be the differentiable reward function of a Markov Decision Process $\mathcal{M}$, $\Pi_{\boldsymbol{\theta}}$ a parametric space of differentiable deterministic policies and $s_t = g(a_{1:t-1})$. Given $\pi_{\boldsymbol{\theta}} \in \Pi_{\boldsymbol{\theta}}$, the* Open-loop Policy Gradient *of $\pi_{\boldsymbol{\theta}}$ under $g$ is defined as:*

$$\nabla_{\boldsymbol{\theta}}^g J(\boldsymbol{\theta}; H) = \sum_{t=1}^H \frac{\partial r(s_t)}{\partial s_t} \sum_{k=1}^{t-1} \frac{\partial g(a_{1:t-1})}{\partial a_k} \frac{\partial \pi_{\boldsymbol{\theta}}(s_k)}{\partial \boldsymbol{\theta}}.$$

This gradient computation resembles the one of the open-loop policy gradient from Definition 1. There, the policy gradient is computed by differentiating through the Markovian dynamics of the environment, while in Definition 2 a gradient is computed by differentiating through an ASM. The next proposition shows how the two definitions are connected, offering a first glimpse at the connection between neural network architectures, ASMs and policy gradient computation in MDPs.

**Proposition 1.** *Let $\mathcal{M}$-RNN be a recurrent network with its recurrent cell being the dynamics $f$ of the Markov Decision Process $\mathcal{M}$. Then,*

$$\nabla_{\boldsymbol{\theta}}^{\mathcal{M}\text{-}RNN} J(\boldsymbol{\theta}; H) = \nabla_{\boldsymbol{\theta}} J(\boldsymbol{\theta}; H).$$

The above proposition tells us that the original policy gradient under an MDP is in fact equivalent to the one computed according to Definition 2, when instantiating $g$ as a recurrent neural network with a specific recurrent cell. Crucially, this not only provides a grounding for gradient-estimation with ASMs, but also hints at a fundamental fact that will be analyzed in-depth in this section: policy gradient computation by differentiation through unrolled Markovian models can be understood to be fundamentally ill-behaved due to its correspondence to an RNN structure. We present in Figure 1 a visual representation of the connection between different neural network architectures used to instantiate an ASM and the resulting computation graph representing the model of the environment.

The sequence modeling perspective provided by the ASM can also be used to interpret architectural and training design choices used in RL. As a concrete example, we present the following remark.

**Remark 1.** *Training an ASM instantiated with an RNN to predict states using teacher forcing is equivalent to training its recurrent cell with the one-step loss function from Equation 1.*

Teacher forcing (Williams & Zipser, 1989) is an algorithm used to train RNNs, that, instead of autoregressively unrolling them during training, enforces the outputs (i.e., in this case, the environment/RNN states) to be the ones coming from a ground-truth trajectory. In other words, when training an ASM as an RNN with teacher forcing, we are essentially training its recurrent cell as a one-step model, as traditionally done in model-based RL applied to a Markovian setting.

## 3.2 Policy Gradient Computation with Action-Sequence Models

Through the concept of action-sequence model, we have established a direct connection between deep sequence models and policy gradient estimation. We will now exploit this connection to characterize the asymptotic behavior of the policy gradient depending on the underlying neural network architecture employed as an ASM. To do so, we leverage the following theorem.

**Theorem 1.** *Let $r$ be the $L_r$-Lipschitz reward function from a Markov Decision Process $\mathcal{M}$, $\Pi_{\boldsymbol{\theta}}$ a parametric space of differentiable deterministic $L_{\pi}$-policies. Given $\pi_{\boldsymbol{\theta}} \in \Pi_{\boldsymbol{\theta}}$, the norm of the open-loop policy gradient $\nabla_{\boldsymbol{\theta}}^g J(\boldsymbol{\theta}; H)$ of $\pi_{\boldsymbol{\theta}}$ under an action-sequence model $g$ as a function of the horizon $H$ can upper bounded as:*

$$\|\nabla_{\boldsymbol{\theta}}^g J(\boldsymbol{\theta}; H)\| \leq L_r L_{\pi} \sum_{t=1}^{H} \sum_{k=1}^{t-1} \left\| \frac{\partial g(a_{1:t-1})}{\partial a_k} \right\|.$$

This theorem holds for any differentiable ASM $g$, and establishes a worst-case relationship between the Jacobian of the ASM w.r.t. its inputs (i.e., actions) and the policy gradient. Notably, the result implies that the policy gradient does not explode if the Jacobian of the ASM does not explode.

We will now leverage the generality of the previous result to characterize the asymptotic behavior of the policy gradient computed using different neural network architectures for an ASM. First, let us consider an ASM instantiated with a simple recurrent neural network with a linear output layer:

$$\text{(RNN)} \qquad x_t = \sigma(W_x x_{t-1}) + W_a a_{t-1} + b; \qquad \hat{s}_t = W_o x_t, \qquad (2)$$

where $\sigma$ is an activation function with gradient norm bounded by $\|diag(\sigma'(x))\| \leq \frac{1}{\beta}$ for some constant $\beta$. Then, the following result holds.

**Corollary 1.1.** *Let RNN be an action-sequence model in the form of Equation 2. For some $\eta > 1$, the asymptotic behavior of the norm of the open-loop policy gradient $\nabla_{\boldsymbol{\theta}}^{RNN} J(\boldsymbol{\theta}; H)$ as a function of the horizon $H$ can be described as:*

$$\left\| \nabla_{\boldsymbol{\theta}}^{RNN} J(\boldsymbol{\theta}; H) \right\| = O\left(\eta^H\right).$$

Corollary 1.1 shows that, in the worst case, the policy gradient computed with an ASM instantiated with an RNN backbone can explode exponentially fast with respect to the problem horizon. As seen in Proposition 1, Remark 1 and Figure 1, using a Markovian model to compute the policy gradient, either given or learned, implies an RNN-structure for the ASM. Thus, Corollary 1.1 explains both the difficulties observed in exploiting differentiable simulators (Metz et al., 2021) and the limitations of existing model-based methods based on differentiating through Markovian models (Heess et al., 2015). This result is a consequence of foundational theory of RNNs (Bengio et al., 1994; Pascanu et al., 2013), demonstrating how the connection created by our ASM framework allows to transfer scientific understanding from sequence modeling to policy gradients.

Let us now apply Theorem 1 to a self-attention ASM (Vaswani et al., 2017). Given $Q \in \mathbb{R}^{1 \times d_z}$, $K \in \mathbb{R}^{n \times d_z}$, $V \in \mathbb{R}^{n \times d_o}$, we define $\text{Attention}(Q, K, V) := \text{softmax}(QK^T)V$. Using a set of weight matrices, a self-attention ASM predicts the next state as follows:

$$\text{(ATT)} \qquad \hat{s}_t = \text{Attention}(a_{t-1} W_q^T, a_{1:t-1} W_k^T, a_{1:t-1} W_v^T). \qquad (3)$$

We can now characterize the behavior of the policy gradient computed with a self-attention ASM.

**Corollary 1.2.** *Let ATT be an attention-based action-sequence model of the form of Equation 3. The asymptotic behavior of the norm of the open-loop policy gradient $\nabla_{\boldsymbol{\theta}}^{ATT} J(\boldsymbol{\theta}; H)$ as a function of the horizon $H$ can be described as:*

$$\left\| \nabla_{\boldsymbol{\theta}}^{ATT} J(\boldsymbol{\theta}; H) \right\| = O\left(H^3\right).$$

This result shows that, inheriting the properties of the underlying sequence model, the policy gradient norm has only a worst-case polynomial dependency on the horizon, instead of an exponential one: by creating a direct connection between the gradient propagation properties of a neural network backbone and the policy gradient, ASMs can take advantage of the better properties of modern architectures based on self-attention to achieve better behaved policy gradients.

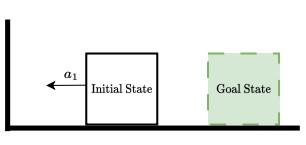 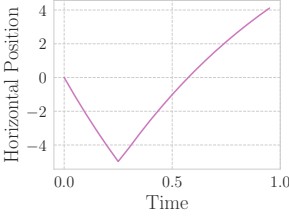 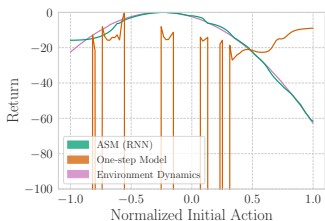

(a) One-bounce environment overview.

(b) Example trajectory non differentiable at $t \approx 0.25$.

(c) Final return with respect to the initial action for different models.

Figure 3: **ASMs ignore non-differentiable points in the state space.** (a) After the block is pushed with some initial action, it bounces off the wall, instantaneously reversing its velocity. (b) Visualization of the point of non-differentiability in the state space. (c) Learning a one-step model is difficult, but an ASM (using an RNN) can still accurately model the final reward when varying the initial action. Learned dynamics are trained offline on a dataset collected using random actions.

## 4 EXPERIMENTS AND DISCUSSION

In this section, we will show that, by casting policy gradients into a sequence modeling problem, the ASM framework unlocks the full potential of modern sequence model architectures for temporal credit assignment in RL. We perform a wide range of experiments to demonstrate that policy optimization with LSTMs or transformers as ASMs architectures outperforms traditional model-free and model-based methods in a variety of domains. In our experiments, we use the differentiable reward function provided by the environment, but we show in Appendix C that ASMs can successfully be employed with learned reward functions as well. We use a Markov agent, or one-step model, to refer to the traditional model-based policy gradient method presented in Section 2, which is a special case of an RNN as highlighted in Proposition 1. When unspecified, we rely on a standard model-based policy optimization paradigm to learn an ASM and use it to optimize a policy, in which we iteratively collect data interacting with the environment and use it to train the model, then compute the policy gradient with the ASM and improve the policy via gradient ascent. We report the pseudocode in Appendix E.

### 4.1 BETTER SEQUENCE MODELS YIELD MORE ACCURATE LONG-TERM GRADIENTS

Section 3 focus on the theoretical stability of policy gradients computed using ASMs. Here, we present complementary empirical evidence that the underlying sequence models can also improve the accuracy of policy gradients in tasks where long-term gradients are well-behaved. We consider the following illustrative *credit assignment* task, inspired by the copy sequence modeling task (Hochreiter & Schmidhuber, 1997). An initial action is taken and copied into the state space for $H$ steps. After $H$ steps, the agent receives a reward as a function of the copied initial action in the final state.

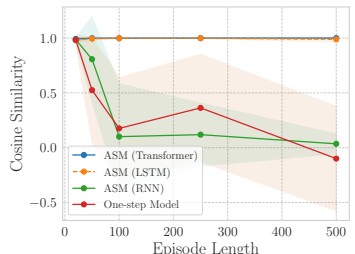

Figure 2: **Better ASMs correspond to more accurate gradients**. Cosine similarity between the policy gradients of different learned dynamics and the true policy gradient on the toy credit-assignment task (10 seeds ± std).

We train ASMs with different architectures on a dataset collected by a randomly-initialized policy, and estimate its policy gradient using the ASMs. We report the cosine similarity between the real policy gradient and the one estimated from the ASMs for different tasks horizons in Figure 2. For large episode lengths, $H$, the policy gradient under the environment dynamics remains stable, yet a traditional Markovian model cannot produce accurate policy gradients for horizons over 100 steps. Gradient estimation with ASMs instantiated using better sequence models instead generates accurate policy gradients for up to 500 steps, showing no signs of degradation. These results demonstrate that better sequence models can directly translate to more accurate policy gradients for RL in the context of our framework.

### 4.2 CAN GRADIENTS FROM MODELS BE BETTER THAN GRADIENTS FROM SIMULATORS?

When having access to a differentiable simulator, it is possible to directly leverage the environment dynamics to compute the policy gradient using Definition 1. However, when the environment

dynamics are not well-behaved, this can lead to unstable optimization, which previous work conjectured might be alleviated by learning neural network models (Metz et al., 2021; Suh et al., 2022). In this section, we show our framework confirms this conjecture, demonstrating that ASMs based on appropriate sequence models can lead to better policy gradients than the ones coming from the environment dynamics, when the simulator features points of non-differentiability or chaotic behavior.

**Partially-differentiable simulators** Real-world systems (e.g., physical systems) often present points of non-differentiability (e.g., contact points) across their state space. However, these points often exists even if the dependency of the rewards from the actions is actually smooth and well-behaved everywhere. In such cases, ASMs can be used to side-step the non-differentiability of the state dynamics, directly mapping from actions to rewards through predicted states. To illustrate this point, we examine a *one-bounce* environment, shown in Figure 3. In this task, an agent must push a block towards a wall with some initial action, such that the block bounces off the wall and ends up at some predetermined goal state after $H$ steps. A single terminal reward is given at $t = H$ measuring the distance of the block to the goal state.

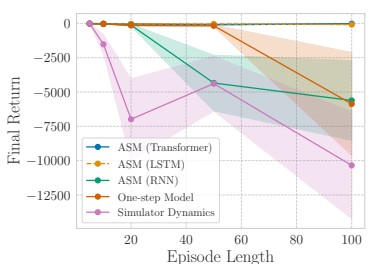

Even though the state trajectory is non-differentiable due to the wall bounce, the final state of the block is actually well-behaved with respect to the permitted initial actions (Figure 3b and Figure 3c). Consequently, we show in Figure 3c that ASMs can accurately predict the final reward, while an unrolled Markovian model cannot make accurate predictions in the presence of points of such abrupt variations in the state space. Notably, the resulting ASM thus allows the use of policy gradient-based optimization even in the presence of an underylying non-differentiable environment dynamics.

Figure 4: **Gradients from ASMs outperform differentiable simulators in chaotic environments.** Performance after policy optimization using different ASMs and the environment dynamics (10 seeds $\pm$ std).

**Chaotic dynamics** Even when a differentiable simulator is available, its dynamics can still be ill-behaved for policy optimization. Chaotic systems constitute a notable example of this, due to the accumulated product of Jacobians leading to gradient explosion (Suh et al., 2022). In Theorem 1, we have shown that an ASM's gradient behavior is determined by its underlying architecture, and explosion may be prevented by the use of an appropriate sequence model. Here, we investigate how these bounds interact with a chaotic environment, and how the resulting policy is affected. To do so, consider a prototypical chaotic system, using a double-pendulum task depicted in Figure 5a. In this environment, the agent's initial action determines the initial angular position of the inner pendulum. Afterwards, the system is rolled out for $H$ steps. The objective is to get the system to be in some predefined goal state after $H$ steps given the initial action. A terminal reward is given which measures the distance between the final observed state and the desired goal state. We illustrate in Figure 5b and Figure 5c the chaotic nature of the task, showing respectively that the norm of true gradient of the final state with respect to the action grows exponentially with

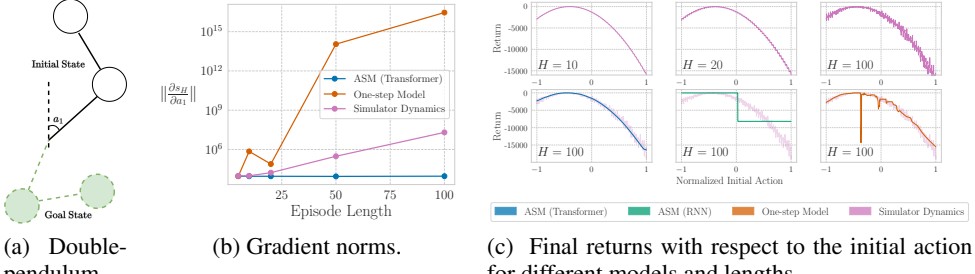

| (a) Double-pendulum. | (b) Gradient norms. | (c) Final returns with respect to the initial action for different models and lengths. |

Figure 5: **ASMs smooths out chaotic dynamics.** (a) A double-pendulum environment where an initial position must be chosen in order to achieve some pre-determined goal state after $H$ steps. Different transition models are learned on a data set of random trajectories. (b) The mean gradient norm of the final state with respect to the initial action for each model is computed over 50 different random actions for each episode length. (c) Final return according to different models for different initial actions and episode lengths $H$.

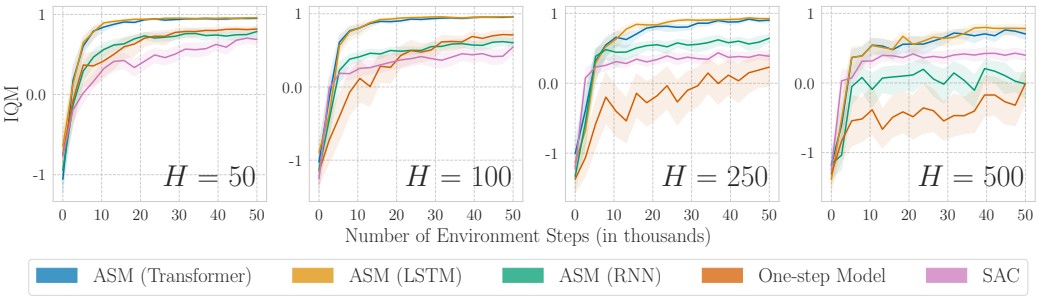

Figure 6: **ASMs are more sample efficient than traditional model-based and model-free methods.** Performance of different ASMs on Myriad, a one-step model, and SAC agent (10 seeds $\pm$ 95% C.I.).

the episode length $H$, and that the return landscape (Rahn et al., 2023) becomes more difficult to navigate as the horizon grows. Figure 5 shows that, while the gradient provided by a one-step model explodes just like the true gradient, a transformer-based ASM provides stable gradients. This manifests itself as a smooth and accurate approximation of the return landscape (Figure 5c), and better policy optimization (Figure 4). These results provide evidence on how compatible with gradient propagation are the inductive biases designed for deep learning architectures, and on how ASMs can take advantage of those inductive biases for policy optimization.

### 4.3 BETTER SEQUENCE MODELS IMPROVE CREDIT ASSIGNMENT

We now show that using advanced sequence models as ASMs results in effective long-term credit assignment. We focus on eight tasks from the Myriad testbed (Howe et al., 2022) inspired by real-world problem (Lenhart & Workman, 2007). We employ Myriad because of its finite-horizon, fully deterministic, continuous and Markovian environments, requiring long-term credit assignment. The episode length, and consequently the credit assignment difficulty, is configurable: reducing the integration step size in an environment effectively increases the horizon $H$. We report additional information in Appendix D. We aggregate results using the IQM (Agarwal et al., 2021).

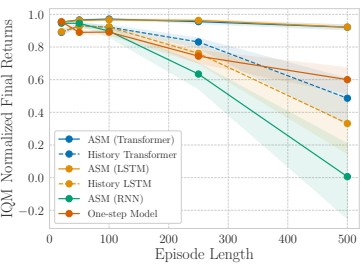

The use of sequence models is common in model-based RL. In fact, many model-based algorithms designed for *partially observable* MDPs use sequence models to approximate its dynamics (Hafner et al., 2022). These history-sequence models resemble ASMs, with an additional conditioning on states, such that $s_t = g(s_1 a_1, \ldots, s_{t-1} a_{t-1})$. While this conditioning may be redundant in a deterministic MDP (in which a sequence of actions implies a single state), it might seem natural to think that such history-sequence models may, in fact, already inherit the benefits of ASMs, given their dependence on the same sequence of actions. To verify this claim, we train an agent following the same policy optimization paradigm but replacing ASMs with history-sequence models.

Figure 7: **Better architectures in ASMs improve long-term credit assignment.** Final performance of policy optimization with different dynamics on Myriad, in the form of ASMs, history-sequence models and one-step models (10 seeds $\pm$ 95% C.I.).

Figure 7 shows the aggregate performance on the Myriad environments. First, ASMs instantiated with an LSTM or a transformer can reliably do well in long-term credit assignment tasks up to a horizon of 500, while an RNN-based ASM and a more traditional one-step model struggle past a horizon of 100. These results corroborate our previous findings: better sequence models can be plugged into the ASM framework to directly improve policy gradients' temporal credit assignment capabilities. Second, we demonstrate that models conditioned on full histories containing both state and actions cannot leverage the inductive biases of the chosen architecture as well as ASMs. Whether an LSTM or a transformer is used, our results show that conditioning on the entire history counteracts the potential benefits of the more powerful sequence models. These results have a simple explanation. Just as history-sequence models process actions as ASMs, they also process state information similarly to a one-step transition function. Therefore, the behavior of policy gradients history-sequence models could be dominated by ill-behaved gradients similar to the ones of unrolled one-step dynamics.

In Figure 6, we also compare ASM-based approaches to the model-free algorithm Soft Actor-Critic (SAC) (Haarnoja et al., 2018). When using transformers or LSTMs, ASMs outperform SAC in sample efficiency. While this is not true for the Markov model-based method, ASMs deliver on the promise of model-based methods for improved sample efficiency over model-free alternatives.

## 5 RELATED WORK

**Differentiating through dynamics and backpropagation.** The concept of learning a policy by differentiating through the environment dynamics has been a foundational aspect of early RL methods (Werbos, 1974; Schmidhuber, 1990). Estimators for such a policy gradient have been employed in different settings and referred to with different names, including pathwise derivatives (Clavera et al., 2020; Hafner et al., 2022; 2023) and value gradients (Fairbank, 2014; Heess et al., 2015; Amos et al., 2021). ASMs connects this literature to the one on sequence modeling with neural networks, with a focus on understanding gradient propagation (Bengio et al., 1994; Hochreiter, 1998; Pascanu et al., 2013) and on architectures that allow for improved credit assignment (Hochreiter & Schmidhuber, 1997; Vaswani et al., 2017; Kerg et al., 2020). Finally, our work can be seen as an instance of gradient-aware model-based RL (D'Oro et al., 2020; D'Oro & Jaskowski, 2020; Abachi et al., 2020), in which we modify the architecture of a model of the dynamics to obtain better policy gradients downstream (Ma et al., 2021).

**Multi-step state prediction with actions.** The idea of training multi-step dynamics, either in the latent space or explicitly, is related to different kinds of previous work in RL. Prior works used models related to ASMs, analyzing their role as partial models (Rezende et al., 2020), in the context of tree search (Schrittwieser et al., 2019), or training them with a multi-step latent prediction loss (Xu et al., 2018; Gregor et al., 2019; Schwarzer et al., 2020), without however ever analyzing their gradient properties. Notably, models used in these prior works are rarely shown to work beyond a few dozen steps of unrolling or imagination, while ASMs demonstrably scale favorably for horizons well beyond established limitations.

**Sequence models in RL.** In partially observable MDPs, sequence models have been extensively used in RL as history encoders to maximize returns with or without world models Hausknecht & Stone (2015); Ni et al. (2021); Hafner et al. (2023). Other works have recently emerged treating MDPs as a sequence modeling problem Chen et al. (2021); Zheng et al. (2022); Janner et al. (2021), showing promise in an imitation learning or offline RL problem setting by deriving policies through return conditioned models. Additional experiments on decision transformers are reported in Appendix F.3. Separately, sequence models have also been used to reshape the reward landscape for improved temporal credit assignment in sparse reward settings Hung et al. (2019); Arjona-Medina et al. (2019); Liu et al. (2019). In contrast to all of these, our framework is the only one to use an *action-only* conditioned sequence model to directly improve long-term policy gradients in MDPs. Appendix G contains a more detailed comparison with all these methods.

## 6 CONCLUSIONS AND LIMITATIONS

In this paper, we presented a differentiable sequence model perspective on policy gradients. We built a framework based on action-sequence models, which are models of the environment dynamics conditioned on sequences of actions, and showed that this framework allows to create a direct connection between sequence modeling with neural networks and policy gradients. In particular, we exploited this connection to theoretically characterize the asymptotic behavior of policy gradients obtained via differentiation through world models. Empirically, we show that better sequence models can be freely plugged into this framework to produce more accurate and well-behaved policy gradients, resulting in more sample efficient and effective long-term credit assignment.

Our work constitutes a first answer to open questions on the relationship of neural network-based dynamics models, sequence models and differentiable simulators. To provide this answer in the most scientifically sound and intelligible form, we opted to use carefully controlled settings and experiments, assuming Markovian and deterministic dynamics, and relatively low-dimensional domains. We encourage future work to leverage our theoretical and empirical insights for scaling up the framework provided by action-sequence models to more complex domains, in stochastic, partially observable, and high-dimensional settings.

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

# Appendix

## Table of Contents

## A PROOFS

**Proposition 1.** *Let* $\mathcal{M}$*-RNN be a recurrent network with its recurrent cell being the dynamics* $f$ *of the Markov Decision Process* $\mathcal{M}$*. Then,*

$$\nabla_{\boldsymbol{\theta}}^{\mathcal{M}\text{-RNN}} J(\boldsymbol{\theta}; H) = \nabla_{\boldsymbol{\theta}} J(\boldsymbol{\theta}; H).$$

*Proof.* We begin by providing a formal definition of the recurrent network with its recurrent cell being the dynamics $f$ of the MDP as:

$$g_t(a_{1:t}) := f(s_t, a_t)$$
$$s_t := f(s_{t-1}, a_{t-1}) = f(g(a_{1:t-1}), a_t)$$

Begin by developing the LHS of the equation.

$$\nabla_{\boldsymbol{\theta}}^{\mathcal{M}\text{-RNN}} J(\boldsymbol{\theta}; H) = \sum_{t=1}^{H} \frac{\partial r(s_t)}{\partial s_t} \sum_{k=1}^{t-1} \frac{\partial g_t}{\partial a_k} \frac{\partial \pi_{\boldsymbol{\theta}}(s_k)}{\partial \boldsymbol{\theta}}$$

$$= \sum_{t=1}^{H} \frac{\partial r(s_t)}{\partial s_t} \sum_{k=1}^{t-1} \frac{\partial s_t}{\partial a_k} \frac{\partial \pi_{\boldsymbol{\theta}}(s_k)}{\partial \boldsymbol{\theta}}$$

where $s_t = f(s_{t-1}, a_{t-1})$ by construction (RNN assumption). Therefore, apply the chain rule $\frac{\partial s_t}{\partial a_k}$:

$$\nabla_{\boldsymbol{\theta}}^{\mathcal{M}\text{-RNN}} J(\boldsymbol{\theta}; H) = \sum_{t=1}^{H} \frac{\partial r(s_t)}{\partial s_t} \sum_{k=1}^{t-1} \frac{\partial s_{k+1}}{\partial a_k} \left( \prod_{i=k+1}^{t-1} \frac{\partial s_{i+1}}{s_i} \right) \frac{\partial \pi_{\boldsymbol{\theta}}(s_k)}{\partial \boldsymbol{\theta}}$$

Now, for the RHS of the equation. Rewriting the policy gradient provided in Definition 1.

$$\nabla_{\boldsymbol{\theta}} J(\boldsymbol{\theta}; H) = \sum_{t=1}^{H} \frac{\partial r(s_t)}{\partial s_t} \sum_{k=1}^{t-1} \frac{\partial s_{k+1}}{\partial a_k} \frac{\partial \pi_{\boldsymbol{\theta}}(s_k)}{\partial \boldsymbol{\theta}} \left( \prod_{i=k+1}^{t-1} \frac{\partial s_{i+1}}{\partial s_i} \right).$$

$$\nabla_{\boldsymbol{\theta}}^{\mathcal{M}\text{-RNN}} J(\boldsymbol{\theta}; H) = \nabla_{\boldsymbol{\theta}} J(\boldsymbol{\theta}; H).$$

$\square$

**Theorem 1.** *Let* $r$ *be the* $L_r$*-Lipschitz reward function from a Markov Decision Process* $\mathcal{M}$*,* $\Pi_{\boldsymbol{\theta}}$ *a parametric space of differentiable deterministic* $L_\pi$*-policies. Given* $\pi_{\boldsymbol{\theta}} \in \Pi_{\boldsymbol{\theta}}$*, the norm of the open-loop policy gradient* $\nabla_{\boldsymbol{\theta}}^g J(\boldsymbol{\theta}; H)$ *of* $\pi_{\boldsymbol{\theta}}$ *under an action-sequence model* $g$ *as a function of the horizon* $H$ *can upper bounded as:*

$$\|\nabla_{\boldsymbol{\theta}}^g J(\boldsymbol{\theta}; H)\| \le L_r L_\pi \sum_{t=1}^{H} \sum_{k=1}^{t-1} \left\| \frac{\partial g(a_{1:t-1})}{\partial a_k} \right\|.$$

*Proof.*

$$\nabla_{\boldsymbol{\theta}}^g J(\boldsymbol{\theta}; H) = \left\| \sum_{t=1}^{H} \frac{\partial r(s_t)}{\partial s_t} \sum_{k=1}^{t-1} \frac{\partial g_t}{\partial a_k} \frac{\partial \pi(s_k)}{\partial \boldsymbol{\theta}} \right\| \le L_r L_\pi \left\| \sum_{t=1}^{H} \sum_{k=1}^{t-1} \frac{\partial g_t}{\partial a_k} \right\| \le L_r L_\pi \sum_{t=1}^{H} \sum_{k=1}^{t-1} \left\| \frac{\partial g_t}{\partial a_k} \right\|.$$

$\square$

**Corollary 1.1.** *Let RNN be an action-sequence model in the form of Equation 2. For some* $\eta > 1$*, the asymptotic behavior of the norm of the open-loop policy gradient* $\nabla_{\boldsymbol{\theta}}^{RNN} J(\boldsymbol{\theta}; H)$ *as a function of the horizon H can be described as:*

$$\left\| \nabla_{\boldsymbol{\theta}}^{RNN} J(\boldsymbol{\theta}; H) \right\| = O\left( \eta^H \right).$$

*Proof.* We begin by defining the transition function of the recurrent network as $x_t = \sigma(W_x x_{t-1}) + W_a a_t + b$, where $\sigma$ is some activation function with gradient norm bounded by $\|diag(\sigma'(x))\| \le \frac{1}{\beta}$.

Further, consider a linear output cell $s_{t+1} = W_o x_t$. Now we begin by showing that $\left\|\frac{\partial x_t}{\partial a_k}\right\| \leq \|W_a^T\|(\|W_x^T\|\frac{1}{\beta})^{t-k}$:

$$\frac{\partial x_t}{\partial a_k} = \frac{\partial x_k}{\partial a_k}\prod_{i=k}^{t-1}\frac{\partial x_{i+1}}{\partial x_i}$$

$$\left\|\frac{\partial x_t}{\partial a_k}\right\| \leq \|W_a^T\|\prod_{i=k}^{t-1}\left\|\frac{\partial x_{i+1}}{\partial x_i}\right\| = \|W_a^T\|\prod_{i=k}^{t-1}\|W_x^T diag(\sigma'(x_i))\|$$

$$\leq \|W_a^T\|\prod_{i=k}^{t-1}\|W_x^T\|\,\|diag(\sigma'(x_i))\|$$

$$\leq \|W_a^T\|\prod_{i=k}^{t-1}\|W_x^T\|\frac{1}{\beta} = \|W_a^T\|(\|W_x^T\|\frac{1}{\beta})^{t-k} \quad.$$

We develop the expression $\frac{\partial s_t}{\partial a_k} = \frac{\partial s_t}{\partial x_{t-1}}\frac{\partial x_{t-1}}{\partial a_k}$, applying the norm and plugging in the above result we obtain:

$$\left\|\frac{\partial s_t}{\partial a_k}\right\| \leq \|W_o\|\,\|W_a^T\|\,(\|W_x^T\|\frac{1}{\beta})^{t-k-1}$$

We then use the bound found in Theorem 1:

$$\|\nabla_{\boldsymbol{\theta}}^g J(\boldsymbol{\theta}; H)\| \leq L_r L_\pi \sum_{t=1}^{H}\sum_{k=1}^{t-1}\left\|\frac{\partial g_t}{\partial a_k}\right\| = L_r L_\pi \sum_{t=1}^{H}\sum_{k=1}^{t-1}\left\|\frac{\partial s_t}{\partial a_k}\right\|$$

$$\leq L_r L_\pi \sum_{t=1}^{H}\sum_{k=1}^{t-1}\|W_o\|\,\|W_a^T\|\,(\|W_x^T\|\frac{1}{\beta})^{t-k-1} = L_r L_\pi \sum_{t=1}^{H}\sum_{k=1}^{t-1}\|W_o\|\,\|W_a^T\|\,\eta^{t-k-1} \quad,$$

where $\eta = \|W_x^T\|\frac{1}{\beta}$. Now we use the assumption that the spectral radius of the recurrent matrix is greater than $\beta$. In the case of the 2-norm, $\eta = \rho(W_x^T)\frac{1}{\beta} > 1$. By this assumption, $\eta^i > \eta^j \ \forall j < i$, therefore we may finalize our proof:

$$\|\nabla_{\boldsymbol{\theta}}^g J(\boldsymbol{\theta}; H)\| \leq L_r L_\pi \sum_{t=1}^{H}\sum_{k=1}^{t-1}\|W_o\|\,\|W_a^T\|\,\eta^{t-k-1}$$

$$\leq L_r L_\pi \frac{H(H+1)}{2}\|W_o\|\,\|W_a^T\|\,\eta^H$$

$$= O(\eta^H).$$

$\square$

**Corollary 1.2.** *Let* `ATT` *be an attention-based action-sequence model of the form of Equation 3. The asymptotic behavior of the norm of the open-loop policy gradient $\nabla_{\boldsymbol{\theta}}^{ATT} J(\boldsymbol{\theta}; H)$ as a function of the horizon $H$ can be described as:*

$$\left\|\nabla_{\boldsymbol{\theta}}^{ATT} J(\boldsymbol{\theta}; H)\right\| = O\left(H^3\right) \quad.$$

*Proof.* We begin with the definition of the self-attention action-sequence model. Let $Q \in \mathbb{R}^{1 \times d_z}$, $K \in \mathbb{R}^{n \times d_z}$, $V \in \mathbb{R}^{n \times d_o}$, attention can be defined as :

$$\text{Attention}(Q, K, V) := \text{softmax}(QK^T)V$$

A self-attention action-sequence model can then be defined in the following way with weight matrices $W_q \in \mathbb{R}^{d_z \times d_a}$, $W_k \in \mathbb{R}^{d_z \times d_a}$, $W_v \in \mathbb{R}^{d_s \times d_a}$.

$$\text{ATT}_t(a_1, ...a_{t-1}) := \text{Attention}(a_{t-1}W_q^T, a_{1:t-1}W_k^T, a_{1:t-1}W_v^T) = \sum_{i=1}^{t-1} c_i(a_iW_v^T) = \hat{s}_t,$$

where $c_i = \text{softmax}_i(a_{t-1}W_q^T W_k a_{1:t-1}^T)$. The subscript on the softmax operator indicates the $i$th index. Now, we begin by showing the expression for $\frac{\partial g_t}{\partial a_k}$:

$$\begin{aligned}
\frac{\partial g_t}{\partial a_k} &= \sum_{i=1}^{t-1} \frac{\partial}{\partial a_k} c_i(a_iW_v^T) \\
&= \sum_{i=1}^{t-1} \frac{\partial c_i}{\partial a_k}(a_iW_v^T) + c_i \frac{\partial(a_iW_v^T)}{\partial a_k} \\
&= c_kW_v^T + \sum_{i=1}^{t-1} \left( c_i(1\{i=k\} - c_k)(a_iW_v^T) \right) \text{ by derivative of softmax } .
\end{aligned}$$

Then, we take the norm:

$$\begin{aligned}
\left\| \frac{\partial g_t}{\partial a_k} \right\| &\leq \|c_kW_v^T\| + \sum_{i=1}^{t-1} \left\| c_i(1\{i=k\} - c_k)(a_iW_v^T) \right\| \\
&\leq \|c_kW_v^T\| + \sum_{i=1}^{t-1} \|c_i(1\{i=k\} - c_k)\| \left\| (a_iW_v^T) \right\| \\
&\leq \|W_v^T\| + \sum_{i=1}^{t-1} \left\| (a_iW_v^T) \right\| \text{ since } |c_i| \leq 1 \ \forall i
\end{aligned}$$

Assuming the actions are bounded by $|a_j| \leq \alpha \ \forall j$,

$$\begin{aligned}
\left\| \frac{\partial g_t}{\partial a_k} \right\| &\leq \|W_v^T\| + \sum_{i=1}^{t-1} \|(a_iW_v^T)\| \\
&\leq \|W_v^T\| + \alpha \sum_{i=1}^{t-1} \|W_v^T\|
\end{aligned}$$

Finally, we use the bound derived in Theorem 1 to finalize the proof:

$$\begin{aligned}
\|\nabla_{\boldsymbol{\theta}}^g J(\boldsymbol{\theta}; H)\| &\leq L_r L_\pi \sum_{t=1}^{H} \sum_{k=1}^{t-1} \left( \|W_v^T\| + \alpha \sum_{i=1}^{t-1} \|W_v^T\| \right) \\
&= O(H^3 \alpha \|W_v^T\|) \\
&= O(H^3).
\end{aligned}$$

$\square$

### A.1 TIGHTNESS OF COROLLARY 1.1

The result of Corollary 1.1 argues that policy gradients through Markovian models may explode as the horizon grows. In order to make this claim, we comment on the tightness of the bound. To do so, consider the line before last in the proof:

$$\|\nabla_{\boldsymbol{\theta}}^g J(\boldsymbol{\theta}; H)\| \leq L_r L_\pi \sum_{t=1}^{H} \sum_{k=1}^{t-1} \|W_o\| \|W_a^T\| \eta^{t-k-1} . \tag{4}$$

To get here, we made use of the result from Theorem 1 which uses a Lipschitz assumption for the reward function and policy and the triangle inequality in its inequalities. The rest of the proof to

get to equation 4 uses the Cauchy-Schwarz inequality to establish all inequalities. The bound in equation 4 is therefore tight; all relevant inequalities can be written as equivalent if and only if all the Jacobians in the recurrent network can be written as a positive scalar multiplication of each other Jacobian, which is a plausible condition. For example, consider the positive one dimensional case where the condition is met. In this case, we may write

$$\|\nabla_{\boldsymbol{\theta}}^g J(\boldsymbol{\theta}; H)\| = L_r L_\pi \sum_{t=1}^{H} \sum_{k=1}^{t-1} \|W_o\| \|W_a^T\| \eta^{t-k-1}$$

$$\geq L_r L_\pi \|W_o\| \|W_a^T\| \eta^{H-1}$$

$$= \Omega(\eta^H) \ ,$$

and the bound is tight.

## B  RELATIONSHIP BETWEEN OPEN AND CLOSED LOOP POLICY GRADIENTS

The policy gradients introduced and used throughout this paper are of an open-loop nature, but the policy itself remains closed-loop. We restate the definition here for convenience.

**Definition 1.**  (Open-loop Policy Gradient under a deterministic MDP). *Let $r$ and $f$ be the differentiable reward and transition functions of a Markov Decision Process $\mathcal{M}$, $\Pi_{\boldsymbol{\theta}}$ a parametric space of differentiable deterministic policies and $s_t = f(s_{t-1}, a_{t-1})$. Given $\pi_{\boldsymbol{\theta}} \in \Pi_{\boldsymbol{\theta}}$, the* Open-loop Policy Gradient *of $\pi_{\boldsymbol{\theta}}$ under $\mathcal{M}$ is defined as:*

$$\nabla_{\boldsymbol{\theta}} J(\boldsymbol{\theta}; H) = \sum_{t=1}^{H} \frac{\partial r(s_t)}{\partial s_t} \sum_{k=1}^{t-1} \frac{\partial f(s_k, a_k)}{\partial a_k} \frac{\partial \pi_{\boldsymbol{\theta}}(s_k)}{\partial \boldsymbol{\theta}} \left( \prod_{i=k+1}^{t-1} \frac{\partial f(s_i, a_i)}{\partial s_i} \right).$$

In the classical reinforcement learning setting, policy gradients are closed loop, and can be derived as the following equations given an MDP (Heess et al., 2015), where $s_t = f(s_{t-1}, a_{t-1})$, and $a_t = \pi_{\boldsymbol{\theta}}(s_t)$:

$$\nabla_{\boldsymbol{\theta}}^* J(\boldsymbol{\theta}; H) := \sum_{t=1}^{H} \nabla_{\boldsymbol{\theta}} r(s_t)$$

$$= \sum_{t=1}^{H} \frac{\partial r(s_t)}{\partial s_t} \frac{ds_t}{d\boldsymbol{\theta}}$$

$$\frac{ds_t}{d\boldsymbol{\theta}} = \frac{\partial s_t}{\partial a_{t-1}} \frac{da_{t-1}}{d\boldsymbol{\theta}} + \left( \frac{\partial s_t}{\partial s_{t-1}} + \frac{\partial s_t}{\partial a_{t-1}} \frac{\partial a_{t-1}}{\partial s_{t-1}} \right) \frac{ds_{t-1}}{d\boldsymbol{\theta}}$$

$$\frac{ds_1}{d\boldsymbol{\theta}} = 0 \ .$$

The open-loop policy gradient defined in Definition 1 is equivalent to the above form when we consider the actions as a constant w.r.t their respective states, or $\frac{\partial a_i}{\partial s_i} = 0$ for all $i$. To fully appreciate this parallel, we can write the closed-loop policy gradient without the recursive dependence, and setting $\frac{\partial a_i}{\partial s_i} = 0$ for all $i$:

$$\nabla_{\boldsymbol{\theta}}^* J(\boldsymbol{\theta}; H) \approx \sum_{t=1}^{H} \frac{\partial r(s_t)}{\partial s_t} \sum_{k=1}^{t-1} \frac{\partial s_{k+1}}{\partial a_k} \frac{\partial \pi(s_k)}{\partial \boldsymbol{\theta}} \left( \prod_{i=k+1}^{t-1} \frac{\partial s_{i+1}}{\partial s_i} \right) \ .$$

With this approximation, which is not unlike truncated backpropagation through time (Elman, 1990; Williams & Zipser, 1995), where hidden states are treated as constants instead, the closed-loop policy gradient is equivalent to the open-loop policy gradient defined in Definition 1.

## C  LEARNING A REWARD FUNCTION

All experiments conducted in this paper assume that the ground truth differentiable reward function is provided. We perform a short ablation on the toy credit-assignment task with and without a learned reward function. Figure 8 shows no significant dip in sample efficiency or performance when the reward function must be learned.

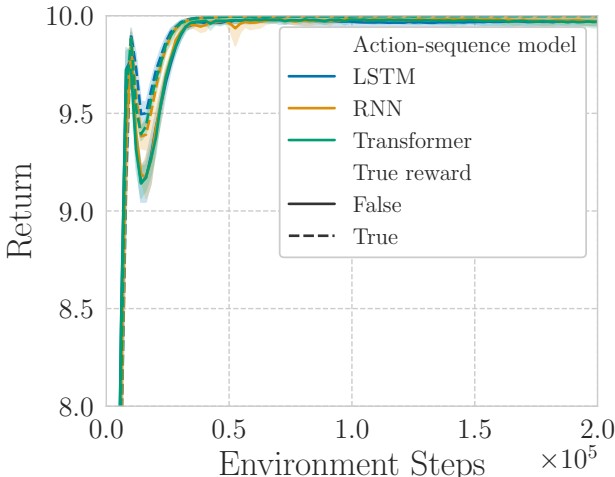

Figure 8: Policy gradients with a learned reward and action-sequence models in the toy credit-assignment task. The horizon is set to 20, and runs are averaged over 10 seeds. The 95% confidence interval is reported.

## D   MYRIAD ENVIRONMENTS

We give a short overview of each of the Myriad environments used in this work. For more details, refer to Howe et al. (2022) and Lenhart & Workman (2007).

The underlying dynamics of each of these environments are described by a set of ordinary differential equations, which are then discretized using Euler's method for discrete-time optimal control. We normalize the returns, where 0 is the expected performance of a random policy and 1 is the performance of an optimal policy provided by Howe et al. (2022) [2]. In total, our experiments are conducted on eight environments: *cancer treatment, bioreactor, mould fungicide, bacteria, harvest, invasive plant, HIV treatment,* and *timber harvest*.

**Cancer treatment** follows the normalized density of a cancerous tumour undergoing chemotherapy. The actions at every time step correspond to the strength of the chemotherapy drug at a given time. The goal is to minimize the size of the tumour over set fixed duration, while also minimizing the amount of drugs administered to the patient.

**Bioreactor** seeks to minimize the total amount of a chemical contaminant that naturally degrades in the presence of a bacteria. The actions here allow the agent to feed the bacteria, increasing its population and increasing the rate of the contaminants degradation. However, a cost is associated to feeding the bacteria.

**Mould fungicide** models the concentration of a mould population. The goal is to minimize its population by applying a fungicide, which has an associated cost to apply.

**Bacteria** looks to maximize a bacteria population through the application of a chemical nutrient that stimulates growth. On top of an associated cost to applying the chemical nutrient, the chemical also produces a byproduct that might in turn hinder bacterial growth.

**Harvest** models the growing population of some vegetable, and the goal is to maximize the harvested yield of this population. While harvesting directly contributes to the reward, it consequently slows down the population's exponential growth.

**Invasive plant** seeks to minimize the presence of an invasive plant species through interventions that remove a proportion of the invasive population. These actions have an associated cost.

---

[2]Optimal policies are computed using trajectory optimization on the underlying differential equations.

**HIV treatment** follows the evolution of uninfected and infected cells in the presence of a virus. The actions correspond to a drug administered that affects the virus' rate of infection. The use of the drug must also be minimized.

**Timber harvest** is similar to the harvest environment, except the harvested population is infinite. Instead, harvested timber can be converted into capital, which can then be re-invested in the harvesting operation, stimulating company growth. The goal is to maximize revenue.

# E  ALGORITHM DETAILS

All model-based RL methods are trained with Algorithm 1. The policies are stochastic and exploration is achieved using maximum entropy with a fixed entropy constant. A deterministic policy is then used for evaluation. All policies are parameterized with a 2-layer 64 hidden units MLP with ReLU activation functions, and all optimization is done using the Adam optimizer (Kingma & Ba, 2014).

---

**Algorithm 1:** Action-sequence Model Policy Gradient Algorithm

---

**Input:** Reward function $r$, ASM $g_\psi$, policy $\pi_\theta$, exploration policy $\pi'$, buffer $\mathcal{D}$
Observe initial state $s_1$ from environment;
**while** *not converged* **do**
    Take action $a_t$ from $\pi'(s_t)$ and observe reward and next state $r_t, s_{t+1}$ from environment;
    Insert transition $(s_t, a_t, r_t, s_{t+1})$ into $\mathcal{D}$;
    Sample $N$ trajectories from $\mathcal{D}$ and update ASM parameters $\psi$ ;
    Update $\theta$ with the policy gradient from Definition 2
**end**

---

In the case of a Markovian agent, the ASM can be initialized as a vanilla RNN with the identity function as its output layer. Training this RNN is also done using teacher forcing to fully recover traditional model-based policy gradient methods such as Heess et al. (2015); Hafner et al. (2022).

## E.1  MODEL-BASED HYPERPARAMETERS

All model-based policy gradient methods, including the Markov agent, ASMs and history-sequence models were swept on three entropy constants for exploration: $[0.1, 0.01, 0.001]$, and the best performing results are reported. All actions, and states (for history-sequence models) are embedded with a linear layer with an output size of 72. The RNN is initialized with two hidden layers, with a hidden layer size of $(64, 64)$. The LSTM also uses two hidden layers of size $(64, 64)$. The self-attention transformers use the architecture for the GPT-2 model (Radford et al., 2019) implemented by the Hugging Face Transformer library (Wolf et al., 2019). Our transformers stack 2 layers and 3 heads of self-attention modules, with hidden layers of size 64. Timesteps are added as an input to every input of all world models, since we are in a finite-horizon setting. The Markov agent models the transition as a difference function: $\hat{s}_t = \hat{f}(s_{t-1}, a_{t-1}) + s_{t-1}$, using two hidden layers of size $(64, 64)$, and ReLU activation functions. Gradient norms are clipped at a value of 100 for all policy gradients as well. All other hyper-parameters relating to the policy learning algorithm are shown in Table 1.

## E.2  MODEL-FREE HYPERPARAMETERS

We use a model-free soft-actor critic Haarnoja et al. (2018) as a benchmark for the myriad experiments. The critic of this agent is modeled as a 2-layer MLP with 256 hidden units each. Entropy regularization is done using a constant entropy constant. Results in Figure 6 show the best performing results after doing a grid search over the following hyper-parameters: learning rate $= [0.001, 0.0001, 0.00001]$ and entropy constant $= [1., 0.01, 0.001]$. The remaining hyperparameters are shown in Table 2.

| Hyperparameter | Value |
|---|---|
| Number of Environment steps | 200000 |
| Dynamics replay ratio | 2 |
| Policy replay ratio | 16 |
| Dynamics batch size | 64 |
| Policy batch size | 16 |
| Dynamics learning rate | 0.001 |
| Policy learning rate | 0.0001 |
| Replay buffer size | 1e6 |
| Warmup steps | 1500 |

Table 1: Hyper-parameters for all model-based algorithms that do not pertain to the world model hyper-parameters.

| Hyperparameter | Value |
|---|---|
| Number of environment steps | 200000 |
| Critic replay ratio | 2 |
| Policy replay ratio | 16 |
| Batch size | 128 |
| Discount | 0.995 |
| $\tau$ | 0.005 |
| Replay buffer size | 1e6 |
| Warmup steps | 1500 |

Table 2: Hyper-parameters for all model-free results ont he Myriad environments.

## F  ADDITIONAL RESULTS AND DETAILS

### F.1  TOY CREDIT-ASSIGNMENT EXPERIMENTS

**Toy Credit-assignment** is an environment designed to test the long-term credit assignment capabilities of an agent in an MDP, therefore separate from its memory capabilities. The state space lies in two dimensions, and the action space is one dimensional. We will use a superscript to denote the dimension of a state. The first dimension is drawn from random distribution $s^1 \sim \mathcal{N}(0, \sigma)$, and the second dimension is 0 at $t = 1$, $s_2^2 = a_1$ and $s_t^2 = s_{t-1}^2$ for $t > 2$. The reward function is given by $r_t = (s_t^1 - a_t)^2/H$ for $t < H$, and $r_t = -20(s_t^2 - 0.5)^2 + 10$. In this paper, to remain deterministic $\sigma = 0$, therefore the optimal course of actions is $a_1^* = 0.5$ and $a_t^* = 0$ for all $t \neq 1$. The maximum return obtained by the optimal agent is 10.

**Offline experiments.** The experiments in Section 4.1 are in the offline setting. In this setting, a random deterministic policy, $\pi_b(s)$, is initialized, and data is collected under this policy with random perturbations, $a_t = \pi_b(s_t) + \mathcal{N}(0, 0.1)$. Specifically, 100000 transitions are collected, and the models are trained for 20000 steps. Afterwards, the policy gradient of $\pi_b$ is computed through the true dynamics, as well the learned dynamics, and their cosine similarity is reported.

**Visualizing Credit Assignment.** Further, we fully train the two layer transformer action-sequence model described above in the online reinforcement learning setting, and visualize the resulting attention weights of the transformer action-sequence model in Figure 9. Indeed, the transformer learns to directly attend to the initial action responsible for all future states in this task, which translates downstream into direct credit assignment from the final reward to the initial action.

### F.2  CHAOTIC EXPERIMENTS

The goal position of the environment is calculated using some fixed initial angular position (initial action) for all episode lengths. In our experiments, the optimal action is -0.4, which corresponds to an initial angle of $-0.4 \times 180°$. The results in the double-pendulum experiments use the same hype-parameters as the Myriad experiments, with the exception that the number of environment steps used for optimization is 100000 instead of 200000.

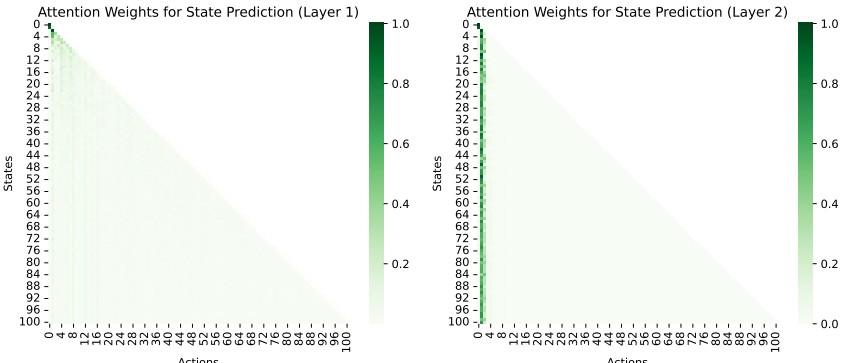

Figure 9: Attention weights for both layers of the transformer action-sequence model trained online on the toy credit assignment task for a horizon of 100 time steps. The $i$th row is the $i$th state prediction, and the $j$th column shows the attention weights to past actions responsible for next state prediction. Weights are averaged over ten random trajectories sampled from the replay buffer. In this case, the second layer is mostly responsible for attending to correct actions.

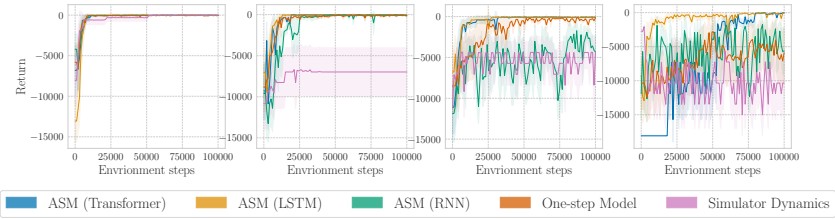

Figure 10: Learning curves of each environment length for the double-pendulum experiments. The lengths presented are $H = [5, 20, 50, 100]$ in order. Ten seeds are reported along with their standard deviations.

### F.3 ONLINE DECISION TRANSFORMER

Recently, a sequence modeling perspective of reinforcement learning has shown promising results on various continuous control tasks in an offline RL setting (Chen et al., 2021; Zheng et al., 2022; Janner et al., 2021). Although our method differs significantly both conceptually and in the problems they solve (see Appendix G), we show experimentally, for completeness, that the online decision transformer (Zheng et al., 2022) performs poorly on the low dimensional Myriad suite. We use the code and hyperparameters provided by Zheng et al. (2022) in a purely online setup, with a sweep on the number of trajectories gathered per iteration due to the online nature of our problem setup. Importantly, decision transformers (DT) must be conditioned on the *return-to-go* (RTG) to derive desired policies. Prior works (Chen et al., 2021; Zheng et al., 2022; Janner et al., 2021) have shown that DTs are robust to this hyperparameters, and can achieve good and sometimes better performance even when the RTG is set to an out of distribution return that is impossible to attain.

We show in Figure 11 that DT performs poorly on the Myriad tasks in an online setting. Figure 12 also shows the individual learning curves of the decision transformer for each task. We summarize some of the takeaways from these experiments:

- Decision transformers are still not well suited for a purely online training regime, exhibiting worse performance than simple one-step model-based methods. Moreover, the action-sequence model outperforms the decision transformer in every single environment.

- Decision transformers require some expert knowledge in many domains, and simply overshooting the return-to-go can yield sub-optimal and sometimes catastrophic results (See Mould Fungicide environment in Figure 12).

- These methods do not scale well with the horizon unlike action-sequence models. Even on relatively short horizons such as 100 time steps, the online decision transformer's performance quickly drops when compared to its performance for 20 time steps.

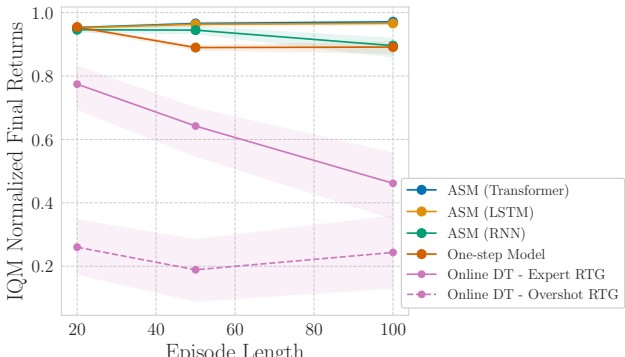

Figure 11: Aggregate performance of an online decision transformer on the Myriad environments. The *Expert RTG* is conditioned on the optimal return for each environment, while the *Overshot RTG* is conditioned on a return about two times higher than the optimal return. All experiments were run with 10 seeds.

### F.4 FINAL PERFORMANCES OF EACH MYRIAD TASK

To the best of our knowledge, this paper presents the first results on using reinforcement learning for the environments in Myriad. The final performances of all methods are shown in Figures 13 and 14.

## G EXTENDED RELATED WORK ON SEQUENCE MODELS IN RL

Sequence models in RL have primarily been used in one of three ways. First, sequence models can be used as history encoders in RL algorithms to maximize returns in partially observable MDPs (POMDPs) (Hausknecht & Stone, 2015; Ni et al., 2021), sometimes through a history-dependent world model (Hafner et al., 2022; 2023). Second, sequence models have recently shown promise in an imitation learning or offline reinforcement learning setting by treating MDPs as a sequence modeling problem (Chen et al., 2021; Zheng et al., 2022; Janner et al., 2021), usually conditioning on returns to derive desired policies. Lastly, a separate line of work have used sequence models to reshape the reward landscape for improved temporal credit assignment (Hung et al., 2019; Arjona-Medina et al., 2019; Liu et al., 2019). In contrast to all of these, our framework is the only one to use an *action-only* conditioned sequence model to directly improve long-term policy gradients in MDPs with no intermediate step. Below, we go into a detailed comparison with each area.

**Comparison with history-conditioned RL methods for POMDPs.** History-conditioned encoders, modeled as sequence models, are often used in POMDPs in both model-free (Hausknecht & Stone, 2015; Ni et al., 2021) and model-based methods (Hafner et al., 2022; 2023). The latter is typically also concerned with predicting observations, they are conditioned on entire history information, while our method includes only actions. The problem setup is also different; such methods are usually concerned with memory in POMDPs (Ni et al., 2023), while ours seeks to improve credit assignment in MDPs.

**Comparison with decision and trajectory transformers.** Decision transformers (Chen et al., 2021; Zheng et al., 2022) and trajectory transformers (Janner et al., 2021) take a more extreme approach, casting the entire reinforcement learning problem as a sequence modeling one. Conversely, we specifically draw a parallel between policy gradients and sequence models. Their sequence models are conditioned on entire trajectories, which include states, actions, rewards and returns. While trajectory transformers predict states just like our action-sequence models, both trajectory and decision transformers also predict actions. In either case, their sequence models must first be trained

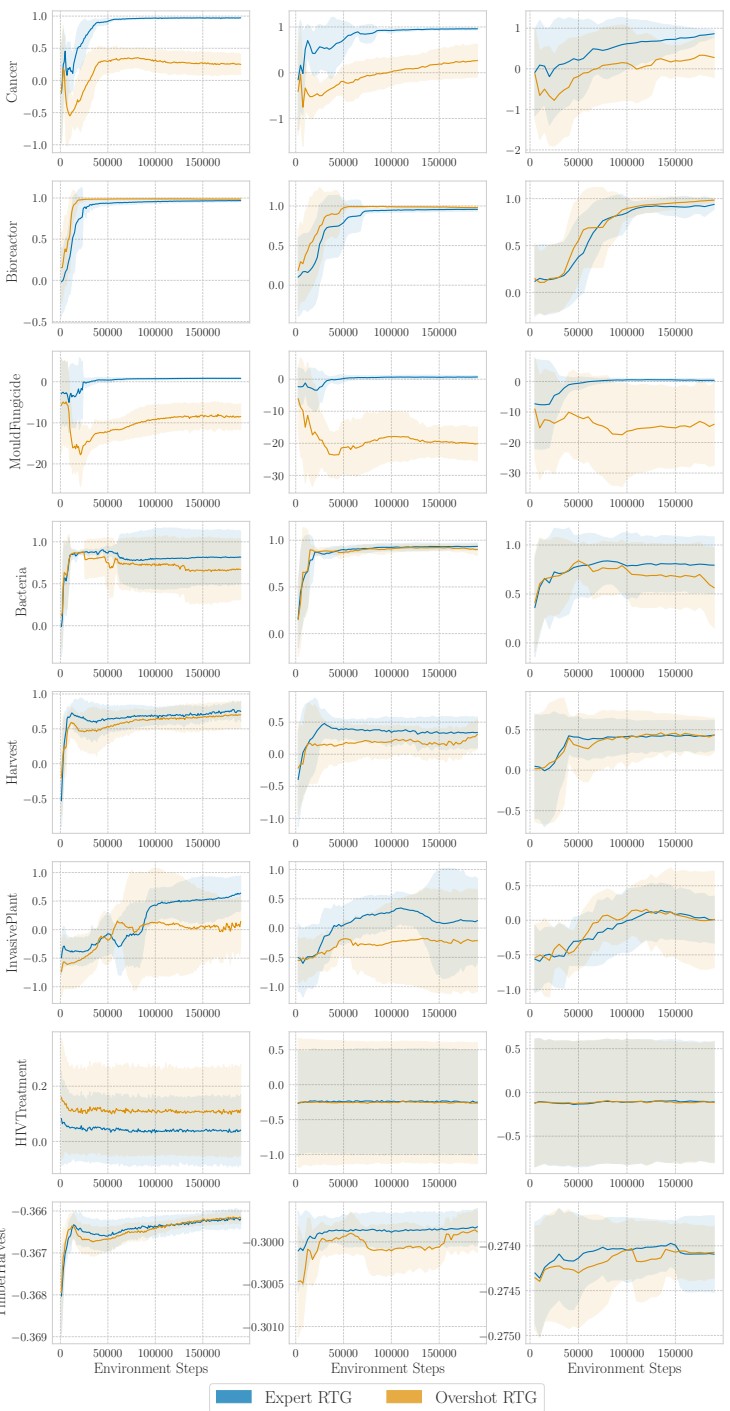

Figure 12: **Learning curves of the online DT for each environment (rows), and horizon (columns, [20, 50, 100])**. The *Expert RTG* is conditioned on the optimal return for each environment, while the *Overshot RTG* is conditioned on a return about two times higher than the optimal return. Each curve shows the mean performance over 10 seeds, and the confidence interval represents the standard deviation.

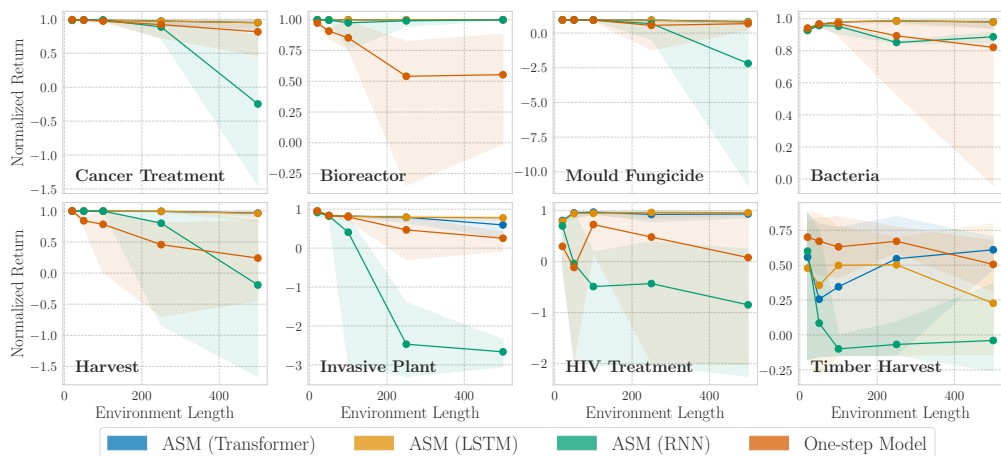

Figure 13: **Better architectures in action-sequence models improve long-term credit assignment.**

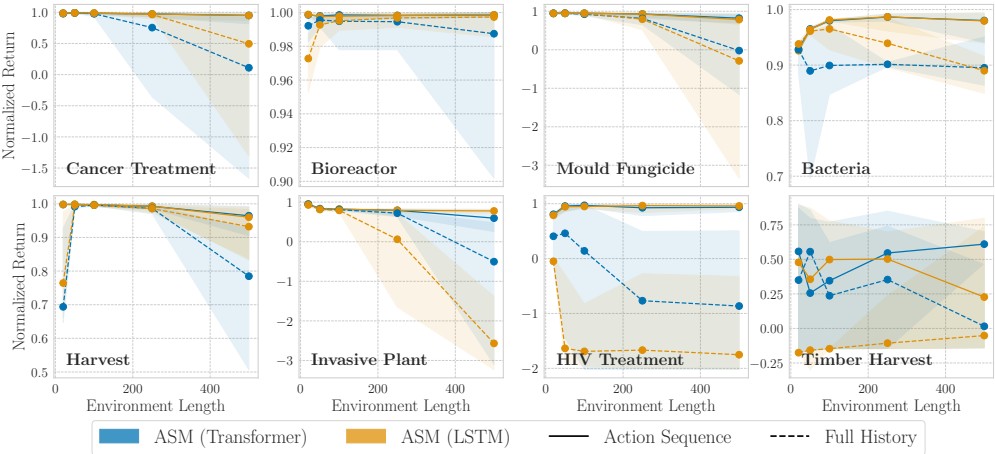

Figure 14: **Less is more: action-sequence models perform better credit assignment than full history dynamics.**

on an offline dataset in an imitation learning (Janner et al., 2021), offline RL (Chen et al., 2021), or pre-training framework (Zheng et al., 2022), while ASMs work for an online RL setting.

**Comparison with reward reshaping methods.** Perhaps more relevant, prior works have already tried harnessing advanced sequence models for improved temporal credit assignment in RL (Hung et al., 2019; Arjona-Medina et al., 2019; Liu et al., 2019). In these cases, the predictive power of sequence models, either an LSTM (Arjona-Medina et al., 2019) or transformer (Hung et al., 2019; Liu et al., 2019) are used to redistribute or augment the given reward function to produce a surrogate reward function. This surrogate reward is then used in a more traditional model-free RL algorithm. Again, we stress a fundamental difference in the inputs of the sequence models, which all include state information. Our framework also establishes a more direct path between sequence models and credit assignment, avoiding any intermediate steps like reward reshaping.

