# OpenReview forum: "A Differentiable Sequence Model Perspective on Policy Gradients"
_ICLR.cc/2024/Conference — Submitted to ICLR 2024_

### Official Review · Reviewer_YpGq · 2023-10-26

**Soundness:** 2 fair
**Presentation:** 3 good
**Contribution:** 2 fair
**Rating:** 3
**Confidence:** 4

**Summary:**

The paper shows a direct connection between backpropagation and policy gradients. The authors thus leverage the advances in deep sequence models to try to improve policy gradient methods. The model proposed is called the Action-Sequence Model (ASM), where the model takes the initial state and the action sequence to predict the state sequence. The authors use a few examples and a testbed called Myriad to demonstrate the effectiveness of the proposed method.

**Strengths:**

The paper tries to study the connection between policy gradient methods and deep sequence models and then improves the stability of the policy gradient methods.

+ The paper provides easy-to-understand illustrations and formulations to show the ideas of connecting deep sequence models and policy gradient methods;
+ The paper conducts experiments with both synthetic tasks like "one-bounce environment", "copy task", etc and real-world tasks like the Myriad testbed.

**Weaknesses:**

- There is some confusion about the experiments, especially on the comparison between action-sequence models and history-sequence models (which have states as conditions). The authors provide some explanations in the final paragraph of Page 8 and Page 9 but I don't think I am convinced. To me, state conditioning is necessary to predict the next states. Only conditioning on the actions does not provide complete information about the environment.

- The connection between policy gradients and RNNs/sequence models seems obvious in the literature. RL policies interact with environments in a recurrent function application manner, which corresponds to RNNs/sequence models. I don't quite see what brings the novel insights from the proposed understanding.

**Questions:**

Please answer and explain the weakness points above.

---

> ### Author Response · Authors · 2023-11-18
> **Authors' Response**
>
> Thank you for your feedback!
>
> > To me, state conditioning is necessary to predict the next states. Only conditioning on the actions does not provide complete information about the environment.
>
> In deterministic environments, which are the object of study of our work, a sequence of actions is always sufficient to exactly determine future states, provided the inital state is given. Empirically, our experimental results show that they are sufficient for effective prediction and credit assignment. Even in certain stochastic environments, action sequences are often sufficient for next state prediction, as is seen from the latent models in [1] [2] and [3].
>
> > The connection between policy gradients and RNNs/sequence models seems obvious in the literature.
>
> We agree that this connection has been intuitively been brought up multiple times, in works we cited in our paper [1] [4] [5]. It can be seen as a contribution of our work to transform this intuitive connection into an actual theoretical and algorithmic framework, that allows to build algorithms with better credit assignment properties by just employing more powerful sequence models.
>
> > I don’t quite see what brings the novel insights from the proposed understanding.
>
> We encourage the reviewer to read our general response and re-evaluate the novelty of our paper. Our proposed understanding allows us, for the first time, to show how any arbitrary advanced sequence model can be used to directly improve credit assignment in policy gradient methods.
>
> [1] Amos, Brandon, et al. "On the model-based stochastic value gradient for continuous reinforcement learning." Learning for Dynamics and Control. PMLR, 2021.
>
> [2] Hafner, Danijar, et al. "Mastering atari with discrete world models." arXiv preprint arXiv:2010.02193 (2020).
>
> [3] Rezende, Danilo J., et al. "Causally correct partial models for reinforcement learning." arXiv preprint arXiv:2002.02836 (2020).
>
> [4] Metz, Luke, et al. "Gradients are not all you need." arXiv preprint arXiv:2111.05803 (2021).
>
> [5] Heess, Nicolas, et al. "Learning continuous control policies by stochastic value gradients." Advances in neural information processing systems 28 (2015).

---

### Official Review · Reviewer_ruP8 · 2023-11-01

**Soundness:** 3 good
**Presentation:** 3 good
**Contribution:** 3 good
**Rating:** 6
**Confidence:** 3

**Summary:**

This paper firstly introduced the task of reinforcement learning and the open-loop policy gradient with a deterministic MDP. Then it reformulates the policy gradient theorem using a sequence (action-sequence) model. By showing these two policy gradients are equivalent, it built the bridge between sequence modeling and the policy gradient.

With the theoretical connection, the authors then demonstrated that it is possible to leverage the advanced network structures in sequence modeling to improve the reinforcement learning, especially the tasks that need temporal credit assignment. This argument is supported with empirical results both under toy experiments and larger scale testbeds.

**Strengths:**

- The paper is well-written and easy to follow. The motivation and the main idea are well presented.
- The experimental results support the claim well, suggesting that advanced sequence modeling/prediction models can indeed lead to better credit assignment prediction.

**Weaknesses:**

I would recommend adding some clarification between the proposed method and various temporal credit assignment methods using sequence models. It would be beneficial to include comparisons in the benchmarks and experiments as well. (Some literature in this domain uses different testbeds, so additional experiments might be necessary.)

**Questions:**

The results would be more convincing if some visualizations of what sequence models have learned are provided. This would help verify that the sequence model is indeed learning the credit assignment property, and that more advanced architectures might indeed enhance performance.

There is literature exploring the use of sequence models for direct temporal credit assignment, such as [1], [2], [3]. It would be beneficial to establish a connection between this work and these references, given the significant overlap in the motivation and methodology. Further clarification of the connections, differences, and novelties would be appreciated.

[1]. Arjona-Medina, J. A., Gillhofer, M., Widrich, M., Unterthiner, T., Brandstetter, J., & Hochreiter, S. (2019). Rudder: Return decomposition for delayed rewards. _Advances in Neural Information Processing Systems_, _32_.

[2]. Hung, C. C., Lillicrap, T., Abramson, J., Wu, Y., Mirza, M., Carnevale, F., ... & Wayne, G. (2019). Optimizing agent behavior over long time scales by transporting value. _Nature communications_, _10_(1), 5223.

[3]. Liu, Y., Luo, Y., Zhong, Y., Chen, X., Liu, Q., & Peng, J. (2019). Sequence modeling of temporal credit assignment for episodic reinforcement learning. _arXiv preprint arXiv:1905.13420_.

---

> ### Author Response · Authors · 2023-11-18
> **Authors' Response**
>
> Thank you for your positive feedback!
>
> > I would recommend adding some clarification between the proposed method and various temporal credit assignment methods using sequence models. [...] It would be beneficial to establish a connection between this work and these references. Further clarification of the connections, differences, and novelties would be appreciated.
>
> We have added to the updated version of our paper a more detailed account of related work. See general response.
>
> > It would be beneficial to include comparisons in the benchmarks and experiments as well
>
> We appreciate this suggestion. Unfortunately, as far as we know, other more common benchmarks in credit assignment are usually POMDPs (at least the reward is history-dependent) and/or discrete in nature [1]. In our paper the main focus is on MDPs with continuous action spaces, where policy optimization via backpropagation through time can be applied. We are excited to generalize our method to such environments in future work.
>
> > The results would be more convincing if some visualizations of what sequence models have learned are provided.
>
> Thanks for the suggestion! We added in Figure 9 of the updated paper a visualization of the attention weights learned by an Action-Sequence Model transfom in the Toy Credit-Assignment problem. For predicting future states, the transformer attends to the only important action, the first one: in this way, when computing the policy gradient, the credit can directly flow from a reward at the end of the sequence to an action at the beginning of the sequence. This happens naturally, just by virtue of backpropagation through time and the particular architecture of the Action-Sequence Model.
>
> [1] Ni, Tianwei, et al. "When do transformers shine in rl? decoupling memory from credit assignment." arXiv preprint arXiv:2307.03864 (2023).

---

> > ### Comment · Reviewer_ruP8 · 2023-11-22
> >
> > I am thankful for the author's response and the subsequent updates made to the paper, particularly the enhanced discussion of related work and the inclusion of similar papers from this domain.

---

> > > ### Author Response · Authors · 2023-11-22
> > >
> > > Thanks for your answer! Let us know if you have any other concerns left. Otherwise, would you consider increasing your score?

---

> > > > ### Comment · Reviewer_ruP8 · 2023-11-22
> > > >
> > > > While I acknowledge the revisions made to the related work section of the paper, unfortunately my current rating remains unchanged. This decision is based on my view that the significance of this work has not been sufficiently clarified to warrant a higher score. Furthermore, the scope of the experiments presented in the paper appears somewhat limited, particularly considering that this work is confined to MDPs with continuous actions only.

---

> > > > > ### Author Response · Authors · 2023-11-22
> > > > >
> > > > > Thank you for the prompt response. We understand your concerns, and hope to address them in the future. Thank you for your feedback!

---

### Official Review · Reviewer_JUws · 2023-11-05

**Soundness:** 2 fair
**Presentation:** 2 fair
**Contribution:** 2 fair
**Rating:** 3
**Confidence:** 4

**Summary:**

The paper endeavors to bridge the gap between gradient propagation in neural networks and policy gradients to advance the field of sequential decision-making. Through theoretical assertions, the paper posits that state-of-the-art neural network architectures can enhance policy gradients. However, the empirical evidence provided to support this claim is not sufficiently convincing, primarily due to the narrow scope of the testing environments utilized. While the authors report improvements in long-term credit assignment and sample efficiency within an optimal control testbed, the paper fails to demonstrate significant innovation or provide a comprehensive comparison with existing methods.

**Strengths:**

The theoretical analysis presented is thorough, suggesting a potential for improved understanding of policy gradient methods.

**Weaknesses:**

1. Originality is a major concern for this submission. The idea of treating RL problems as sequence modeling tasks is not new and has been extensively covered in prior work, specifically in [1] and [2]. This paper does not clearly establish its unique contributions to the field, and the related work section is insufficiently detailed, lacking a critical analysis of how this work diverges from existing methodologies.

2. The experimental design does not effectively differentiate the proposed method from established sequence modeling algorithms. A more robust comparison to state-of-the-art sequence modeling techniques, while not SAC and One-Step model, is necessary to validate the claims of the paper. Additionally, the benchmarks chosen for testing the methodology do not cover the breadth of scenarios needed to substantiate the authors' assertions. The inclusion of common continuous control benchmark tasks, such as Hopper, HalfCheetah, Walker and Antmaze, is essential for a more comprehensive evaluation.

[1] Janner, Michael, Qiyang Li, and Sergey Levine. "Offline reinforcement learning as one big sequence modeling problem." Advances in neural information processing systems 34 (2021): 1273-1286.

[2] Chen, Lili, et al. "Decision transformer: Reinforcement learning via sequence modeling." Advances in neural information processing systems 34 (2021): 15084-15097.

**Questions:**

See weaknesses.

---

> ### Author Response · Authors · 2023-11-18
> **Authors' Response**
>
> Thank you for your feedback!
>
> > The idea of treating RL problems as sequence modeling tasks is not new and has been extensively covered in prior work, specifically in [1] and [2]
>
> Our work _does not treat the RL problem as whole as a sequence modeling task_. Instead, it is geared towards a problem formulation and general methodology based on policy gradients, and relies on sequence modeling as a tool to learn a model of the dynamics, and to connect the theory of deep learning to RL. Indeed, a large part of the conceptual appeal of our proposed framework compared to e.g., decision transformers, is that we are able to leverage powerful sequence models, at the same time being grounded in the traditional theory of model-based policy gradients. To state these differences in an even clearer way, in addition to the discussion about the work mentioned by the reviewer already present in our submission, we have included in the updated paper a detailed account of the relationship between our work and other work employing sequence models in RL.
>
> >  This paper does not clearly establish its unique contributions to the field, and the related work section is insufficiently detailed.
>
> The length of our related work section was only due to space constraints. We included more discussion on the related work in the main text, the appendix of the updated paper, and on the general response above.
>
> > The experimental design does not effectively differentiate the proposed method from established sequence modeling algorithms.
>
> Our method does not introduce any new "sequence modeling algorithm". As opposed to that, the rationale behind our work is to marry together well-established deep learning architectures and training techniques with the core RL machinery of policy gradients. The result is a framework that, in its simplicity, both provides new theoretical understanding and natural ways to get improved credit assignment. We believe our work is a first stepping stone exploiting advanced sequence models in policy gradient methods, and we hope future work will put engineering efforts to scale our methodology to more complex tasks.
>
> > A more robust comparison to state-of-the-art sequence modeling techniques, while not SAC and One-Step model, is necessary to validate the claims of the paper.
>
> We are not aware of any state-of-the-art sequence modeling RL algorithms immediately appropriate for our problem setting, which is concerned with long-term credit assignment in deterministic MDPs for online RL. The closest method highlighted in this review is the online decision transformer from [1]. We are in the process of running experiments using online decision transformers applied completely online, and will provide an update as soon as the results are out (**See edit, the experiments we're added into Appendix F.3**).
>
>
> > The inclusion of common continuous control benchmark tasks, such as Hopper, HalfCheetah, Walker and Antmaze, is essential for a more comprehensive evaluation.
>
> Our method is empirically evaluated for its ability to solve difficult temporal credit assignment tasks. As shown by recent work [2], the mentioned robotics locomotion benchmarks are not interesting from a temporal credit assignment perspective, and we are not aware of any common continuous temporal credit assignment MDPs beyond the environments we used. Also, note that, despite being generally low-dimensional, tasks from Myriad are based on existing dynamical systems and realistic applications such as cancer treatment, and could be considered by some readers as compelling as robotics locomotion, if not more.
>
> [1] Zheng, Qinqing, Amy Zhang, and Aditya Grover. "Online decision transformer." international conference on machine learning. PMLR, 2022.
>
> [2] Ni, Tianwei, et al. "When do transformers shine in rl? decoupling memory from credit assignment." arXiv preprint arXiv:2307.03864 (2023).

---

> ### Author Response · Authors · 2023-11-20
> **Additional Results on Decision Transformers**
>
> **Edit 20/11/2023**: We have included additional results on the online decision transformer on the Myriad environments. Please see Appendix F.3 and the general response. We hope that these results, along with our extended related work, clears up any confusion that our work does not differ enough from the line of work on decision transformers.

---

### Official Review · Reviewer_KDm5 · 2023-11-06

**Soundness:** 2 fair
**Presentation:** 2 fair
**Contribution:** 2 fair
**Rating:** 5
**Confidence:** 3

**Summary:**

The paper proposes a model-based deterministic policy gradient method for finite-horizon MDPs with deterministic and differentiable transition kernel. In this case the cumulative reward is deterministically determined given a deterministic policy, and its gradient with respect to the policy parameters, i.e. the policy gradient, can be computed by differentiating the transition kernel. Since the true transition kernel is unknown, a baseline solution is to learn a one-step Markovian transition model, then computing the policy gradient by differentiating this learned Markovian transition model. This paper however proposes to not learn one-step model, but to learn a multi-step transition model which takes a sequence of actions as input and predicts a sequence of resulted states as output. Such a multi-step transition model is called Action Sequence Model (ASM) in this paper. The policy gradient is then computed by differentiating over the learned ASM. Despite the Markovian property of the true transition kernel, the paper argues that learning such an multi-step ASM model is a better choice than learning a one-step Markovian model, for the sake of gradient based policy optimization.

It should be noted that the policy gradient discussed in this paper is limited to "open-loop policies" that generate actions without taking the observed states into account. For the more general class of close-loop policies,  the so-called "open-loop policy gradient" as defined and discussed in this paper is not the true policy gradient, but is related to the true gradient in the sense that we can obtain the former if ignoring the $\partial a/\partial s$ terms in the latter.

**Strengths:**

In general, I think it is an very interesting topic to explore whether in model-based RL it's benefitable to learn not directly the underlying MDP model but another form of model. The experiment part of the paper made several good points to this end. For example, the Chaotic experiment in Section 4.2 nicely illustrates a case where even perfectly learning the Markovian transition model can lead to chaotic policy gradients while ASMs can smooth out the gradients and therefore lead to better policy optimization. It is also quite intriguing to see that, in Section 4.3, ASMs lead to better policy optimization than models with full history info including the states (although I'm not sure about the explanation about this phenomenon provided in the paper).

**Weaknesses:**

**(a)** I am not sure that the theory part (Section 3) well support the claim that action-sequence models are better choice than one-step Markovian models. Only the norm of the gradients induced by two special cases of these two classes of models are compared, but in practice the estimated gradient is often normalized so the norm is less important, in my impression. On the other hand, the accuracy of the gradient direction may be a more important factor, I suspect, but is not analyzed at all in the theory part. Also see my Question 1~4 below for several soundness concerns about the theory part.

**(b)** I am not sure if the baselines in the experiment part (Section 4) are strong enough to establish the advantage of ASM over one-step models. In particular, the one-step model tested in the experiment seems to be a very simple one. See my Question 5~6 below for the detailed concerns.

**(c)** The current results of this paper seem to have limited applicability scope: it seems to mainly applicable to environment that is deterministic, differentiable, with fixed episode length, where open-loop policies are sufficient for the environment. In this special case, the RL problem degenerates to a simple black-box optimization problem where we maximize an unknown but deterministic objective function over an action-sequence space. It would be more interesting if the paper can discuss more complex situations, such as those that require close-loop control.

**Questions:**

1. Page 4, in the paragraph below Proposition 1, you said it's a "fundamental fact" that PG with Markovian model is "fundamentally ill-behaved", and you said this fact is "analyzed in-depth in this section". I don't quite understand this sentence and am not sure about the analysis either. By "ill-behaved" do you mean the exponential upper bound in Corollary 1.1? But Corollary 1.1 applies only to a special Markovian model, the model with linear units. What about other Markovian models? In what sense can we conclude that Corollary 1.1 is due to the Markovian property, instead of to the linearity or other limits of the model under consideration?

2. How do we know that the upper bound in Corollary 1.1 is tight? Without tightness, an upper bound like Corollary 1.1 is not enough to support your claim that the gradient of RNN models will "explode exponentially fast". To support such a claim, we typically need a *lower bound* result.

3. Even though it really could be proved that the gradient of Markovian model with true transition kernel grows exponentially with the horizon length -- even though we suppose this were true in this question -- this means the *unbiased* policy gradient optimization is unstable and we perhaps should not use the model-based policy gradient method at all in this case, isn't it? Importantly, although the gradient with Transformer is better bounded in this case, since we know that it's *not* the true policy gradient (because the true policy gradient has larger norm), how do we know that the gradient from transformer is different from an arbitrary small-but-biased gradient, in terms of its effectiveness to power the policy optimization?

4. Page 5, in the paragraph below Corollary 1.1, you said "Corollary 1.1 explains both the difficulties ... and the limitations ...". What are exactly the difficulties and limitations here? Why does your upper bound result indeed *explain* them (rather than just coincident with them)? The argument here is not self-contained so it's hard to evaluate its soundness.

5. In your experiments, what's the difference between the "ASM(RNN)" model and the "One-step Model"? Are they use the same linear transition kernel given by Eq.2, that is, is ASM(RNN) equivalent to unrolling one-step model for H steps? While you upgrade the ASM models from simple RNN to LSTM and Transformer, did you also try to upgrade the one-step model from a simple linear model to more sophisticated ones?

6. In Section 4.3, are the environments here partially observable? Does a state info $s_t$ give a Markovian state or only the partial observation of the full state? I am not sure that the capability to see the additional state info for "History Transformer" can really account for its bad performance, given that the attention modules in Transformer can be trained to simply ignore the state info if they are not helpful. On the other hand, one-step Markovian models are just not appropriate for POMDP environments, so I'm not sure they should be included as baselines in this experiment.

---

> ### Author Response · Authors · 2023-11-18
> **Authors' Response**
>
> Thank you for the detailed feedback, we appreciate the thoroughness of your review!
>
> > in practice the estimated gradient is often normalized so the norm is less important, in my impression.
>
> Clipping exploding gradients often results in biased gradients and does not always provide a benefit [1]. In fact, we tried to normalize gradients for all of our experiments, for one-step and true models, via both clipping and normalization, with no signficant gain in performance.
>
> >   On the other hand, the accuracy of the gradient direction may be a more important factor, I suspect, but is not analyzed at all in the theory part.
>
> Indeed, for some environments, accuracy can play an important role, but it is difficult to analyze in theory, since it would require accurately characterizing the training dynamics and generalization of a neural network. Instead, we provide empirical evidence to complement and confirm our theory: experiments in Section 4.1 show improved gradient accuracy for well-behaved systems, and the double-pendulum experiment in Section 4.2 demonstrates a case where using the gradient coming from a smooth but possibly inaccurate model leads to better policy optimization compared to using the real policy gradient.
>
> This highlights a highly desirable property of using architectures such as Transformers as Action-Sequence Models. On the one hand, they are able to provide an accurate gradient when the dynamics of the environment is smooth enough; on the other hand, when the dynamics is non-smooth and thus not amenable to policy optimization, they will tend to approximate it with a smooth version of it, possibly surpassing the original dynamics in terms of ease of optimization via backpropagation through time.
>
> >  It would be more interesting if the paper can discuss more complex situations, such as those that require close-loop control.
>
> Note that our current analysis and experiments employ a closed-loop **policy**, but simply use an open-loop **policy gradient**, identical to the gradient used in state of the art model-based methods such as [2], [3] and [4]. The open-loop **policy gradient** does not necessarily induce an open-loop **policy**, and, despite its effect is yet to be well understood in the literature, it has been shown to lead to successful policies in complex tasks.
>
> > By “ill-behaved” do you mean the exponential upper bound in Corollary 1.1?
>
> Yes, Corollary 1.1 formally establishes the possibility of exploding gradients under a Markovian model.
>
> > But Corollary 1.1 applies only to a special Markovian model, the model with linear units. What about other Markovian models?
>
> In reality, the analysis provided by Corollary 1.1 considers a non-linear network $x_t=\sigma(W_x x_{t-1}) + W_a a_{t-1} + b$. The linearity with respect to the action quickly becomes non-linear for all future state predictions due to the recursive nature of $x_t$. The analysis can thus be extended to the more traditional non-linear formulation $x_t = \sigma(W_x x_{t-1} + W_a a_{t-1} + b)$, just as done in the seminal work by Pascanu [5].
>
> > Without tightness, an upper bound like Corollary 1.1 is not enough to support your claim that the gradient of RNN models will "explode exponentially fast".
>
> In our paper, we only claim "*the policy gradient [...] with an RNN [...] **can** explode exponentially fast [...]*".
>
> We agree that a tight bound is important in order for our statement to be relevant: for this reason, we have added a justification in Appendix A.1, explaining why the bound is indeed tight. The justification is similar to the ones used to explain exploding gradients in RNNs in supervised learning [1] [5].
>
> > Even though it really could be proved that the gradient of Markovian model with true transition kernel grows exponentially with the horizon length, [...] this means the unbiased policy gradient optimization is unstable and we perhaps should not use the model-based policy gradient method at all in this case.
>
> This is a good observation. However, the fact that the policy gradient under the true transition kernel does not lead to successful policy optimization does not imply, surprisingly, that we should not use model-based policy gradients at all. Our experiments on the double-pendulum environment in Section 4.2 exactly demonstrate this point: the learned gradient from a better sequence model is highly preferable to the true gradient, and amenable to policy optimization.
>
> > how do we know that the gradient from transformer is different from an arbitrary small-but-biased gradient, in terms of its effectiveness to power the policy optimization
>
> While we cannot prove this theoretically, we show empirically in Section 4.2 that the learned well-behaved gradients from a Transformer are indeed effective for policy optimization, leading to high-perfoming policies compared to other baselines.

---

> ### Author Response · Authors · 2023-11-18
> **Authors' Response (cont.)**
>
> > you said “Corollary 1.1 explains both the difficulties … and the limitations …”. What exactly are the difficulties and limitations here?
>
> Prior works in model-based policy gradients have noted the difficulties of unrolling models for long horizons, and Corollary 1.1 justifies this difficulty due to the exploding gradients.
>
> > Why does your upper bound result indeed explain them (rather than just coincide with them)?
>
> These exploding gradients are subsequently shown in Figure 5 to result in poor policies in Figure 4.
>
> > In your experiments, what's the difference between the "ASM(RNN)" model and the "One-step Model"?
>
> Fundamentally, we note in Section 3.1 that they differ only in their training objectives: *"When training an ASM as an RNN with teacher forcing, we are essentially training its recurrent cell as a one-step model"*. Experimentally, the ASM(RNN) is detailed in Appendix E: it is modeled as a 2-layer RNN with ReLU activations. Notably, we have added the architecture details of the one-step model, which also uses a 2-layer MLP with non-linear ReLU activations.
>
> > did you also try to upgrade the one-step model from a simple linear model to more sophisticated ones
>
> Yes, all experiments use a non-linear 2 layer MLP for the one-step model. No experiment uses one-step linear models.
>
> > In Section 4.3, [...] are the environments here partially observable? [...] one-step Markovian models are just not appropriate for POMDP environments.
>
> No, as specified in our background section, all the environments we employ in our paper are fully-observable and Markovian.
>
> > I am not sure that the capability to see the additional state info for "History Transformer" can really account for its bad performance, given that the attention modules in Transformer can be trained to simply ignore the state info if they are not helpful.
>
> The poor performance of the History Transformer does not come from the inability of a transformer to ignore uninformative state information. Instead, for most environments, a transformer conditioned on the entire history will find the state information particularly useful for predicting the next states. As a notable example, in a Markovian case, the transformer can actually learn to just use the last state and action to predict the next state: in this case, unrolling the transformer will create a computational graph that would be similar to the one of an unrolled Markovian model, and thus potentially lead to ill-behaved gradients. For its particular structure, an Action-Sequence Model is instead constrained to directly attend to the actions at its input, thus directly propagating gradients from rewards to actions without unnecessarily chaining next-state predictions.
>
>
> [1] Metz, Luke, et al. "Gradients are not all you need." arXiv preprint arXiv:2111.05803 (2021).
>
> [2] Hafner, Danijar, et al. "Mastering atari with discrete world models." arXiv preprint arXiv:2010.02193 (2020).
>
> [3] Hafner, Danijar, et al. "Mastering diverse domains through world models." arXiv preprint arXiv:2301.04104 (2023).
>
> [4] Ghugare, Raj, et al. "Simplifying model-based rl: learning representations, latent-space models, and policies with one objective." arXiv preprint arXiv:2209.08466 (2022).
>
> [5] Pascanu, Razvan, Tomas Mikolov, and Yoshua Bengio. "On the difficulty of training recurrent neural networks." International conference on machine learning. Pmlr, 2013.
>
> [6] Heess, Nicolas, et al. "Learning continuous control policies by stochastic value gradients." Advances in neural information processing systems 28 (2015).

---

> > ### Comment · Reviewer_KDm5 · 2023-12-02
> > **Response to the authors' rebuttal**
> >
> > **Regarding weakness (c)**
> >
> > > It would be more interesting if the paper can discuss more complex situations, such as those that require close-loop control.
> >
> > Note that in the above sentence I was not commenting on the type of *policy* that your paper has dealt with; instead, I was commenting on the type of *environment* that your paper is considering. Regardless of whether the paper has studied open-loop or close-loop policies, open-loop policies are enough for the type of environments assumed here which are fully deterministic -- even the episode length and the initial state are fixed.
> >
> > The open-loop policy gradient vs close-loop policy issue that your rebuttal is talking about corresponds to my remark in the second paragraph of the Summary section of my review. This is an another issue. Regarding this issue, I acknowledge that people are free to use the so-called "open-loop policy gradient" formula to update a close-loop policy, but it's better to not call the formula "policy gradient" in this case, in my opinion, because it is really *not* the gradient of the episodic return with respect to the policy parameters. The "approximation" required to turn the true policy gradient into this formula is really worrisome -- we need to ignore the fact that the change of state can lead to change of action which is literally what close-loop control is all about. Even if the formula worked in practice -- I'm not sure about this in general cases but even if this were true -- it's still not clear if it were working *as (an approximation of) the policy gradient*.
> >
> > **Regarding weakness (a) and Question 1-4**
> >
> > I now realize a good point after reading your rebuttal, although I'm not sure it's indeed what you meant: Specifically, now I see that Corollary 1.1 demonstrates a fundamental limit of the *true* policy gradient for a certain type of Markovian *environment* -- suppose in the *true* MDP the *true* Markovian transition function of the environment is of the form of a linear RNN, then the *true* policy gradient can be exploding, and this is regardless of what kind of neural network models an RL agent in that environment is potentially using to model/approximate this true transition kernel.
> >
> > I'm not sure that the current Corollary 1.1, and Section 3.2 in general, is indeed reflecting the above perspective -- in particular, Corollary 1.1 in the current draft is still talking about a certain ASM which is a *model* of the environment rather than the environment itself. I encourage the authors to keep revising this part of the paper.
> >
> > On the other hand, the rebuttal seems to acknowledge that the current theoretical results (even with the above perspective) are a bit limited, leaving many important points, such as the "smoothing out" effect, illustrated by experiments only. In this case, I suggest the authors to consider making a less formal discussion in Section 3, focusing more on delivering insights and articulating hypothesis that are to be examined by the experiments later.
> >
> > **Regarding weakness (b) and Question 5-6**
> >
> > Thanks for the clarifications, but my concern about the strength of the one-step model, the baseline, mostly remains. In particular, in Q5 I'm asking the *difference* between ASM(RNN) and One-Step Model in your experiment. Your rebuttal seems to have re-iterated the *similarity* between the two. You said "they differ only in their training objectives", but what are their respective training objectives then? The sentence you cited is not giving the objective functions. I thought they both use the MSE loss as the training objective function, as Eq.(1) in your paper gives, no?
> >
> > Also, in Appendix E it's only said "All *policies* are parameterized with a 2-layer 64 hidden units MLP with ReLU activation functions", and your rebuttal seems to re-iterate this sentence. But what I'm questioning about is the architecture of the learned *transition model* (for ASM-RNN and One-Step-Model, respectively), not the architecture of the policy model.
> >
> > And regarding Q6, I'm not sure the rebuttal has well addressed the question here, but since this is mostly an your-intuition-vs-my-intuition debate, without solid evidence from either side, I'm less motivated to follow up on this issue and tend to consider it as a minor concern *as long as* the counter-intuitive observation about History-Transformer here is indeed true, general, and reproducible.

---

> > ### Comment · Reviewer_KDm5 · 2023-12-02
> > **Additional Notes**
> >
> > I notice that other reviewers have concern on the novelty of the "sequence model perspective", which is highlighted in the paper title even. I actually had the same concern when reading the paper title, which could make many people immediately recall the Decision Transformer work, I'm afraid.
> >
> > Regarding this, I invite the authors to consider the possibility to re-position the paper a little bit. Perhaps the general "perspective" in this paper is indeed not entirely novel -- in fact, I'm not sure that it's even fair to attribute this perspective to the Decision Transformer paper either, as the "Upside-Down RL" work by (Juergen Schmidhuber, 2019) seems to have proposed similar perspective earlier, for example. In any case, leaving aside the novelty at the perspective level, I found this paper more interesting at the "approach" level, in terms of how to use a sequence model, and what kind of sequence model architecture is better, in the context of model-based RL.

---

### Author Response · Authors · 2023-11-18
**General Response**

We thank the reviewers for their valuable feedback. Overall, the reviewers noted that the paper was easy to follow and understand. The experiments shed valuable and intriguing insight into the use sequence models for model-based policy gradients, as noted by a few of the reviewers (KDm5, ruP8, YpGq). This, along with theoretical contributions (JUws), open the door towards future work in adopting deep sequence models towards efficient policy gradients in RL.

A primary concern raised by almost all reviewers (JUws, ruP8, YpGq) is in how our work is positioned amongst the literature of sequence models in RL. To adrress this issue, we have amended related works  and the Appendix to include a detailed comparison of our work with respect to others involving sequence models in RL. To further clarify the unique qualities of our research, we restate here some of the main contributions:
- We show, for the first time, that better neural networks architectures directly corresponds to better policy gradients both in theory and in practice.
- We provide a framework that formalizes the intuitions from recent work on the connection between model-based policy gradients and gradient propagation in sequence models, proposing a method to overcome the limitations (of long horizon model-based policy gradients) that have been previously highlighted [3] [4] and advancing the scientific understanding of this problem.
- We provide experimental answers to the *open question* from the recent policy gradient literature, of whether artificial neural networks can provide better policy gradients than the real differentiable simulator, in the case of stiff or chaotic dynamics [5].

We summarize here as well the main differences of our work with other sequence models in RL, but encourage the reader to refer to the amended Appendix and related works:
- **History-conditioned RL for POMDPs.** Our sequence models are mainly conditioned on actions, and focus on improving credit assignment while history-conditioned RL mainly addresses memory, which are two distinct problems [1]. An empirical comparison is also provided in Section 4.3.
- **Decision and trajectory transformers.** These methods have only been studied in an offline RL, imitation learning, or pre-training setting [2]. Their models are conditioned on entire trajectories and differ completely from traditional policy gradients. In stark contrast, our framework works in an online RL setting, taking both a theoretical and empirical **gradient** perspective, while conditioning only on actions.
- **Reward reshaping with sequence models.** Methods such as [6] [7] [8] share one thing in common with our work: improving temporal credit assignment using sequence models. However, these methods all use state-conditioned sequence models in order to augment or completely replace a given reward function, which is then used in a standard model-free algorithm. Our work directly improves credit assignment through the gradient properties of better sequence models, avoiding any intermediate steps like reward shaping.




[1] Ni, Tianwei, et al. "When do transformers shine in rl? decoupling memory from credit assignment." arXiv preprint arXiv:2307.03864 (2023).

[2] Zheng, Qinqing, Amy Zhang, and Aditya Grover. "Online decision transformer." international conference on machine learning. PMLR, 2022

[3] Metz, Luke, et al. "Gradients are not all you need." arXiv preprint arXiv:2111.05803 (2021).

[4] Heess, Nicolas, et al. "Learning continuous control policies by stochastic value gradients." Advances in neural information processing systems 28 (2015).

[5] Suh, Hyung Ju, et al. "Do differentiable simulators give better policy gradients?." International Conference on Machine Learning. PMLR, 2022.

[6] Arjona-Medina, J. A., Gillhofer, M., Widrich, M., Unterthiner, T., Brandstetter, J., & Hochreiter, S. (2019). Rudder: Return decomposition for delayed rewards. Advances in Neural Information Processing Systems, 32.

[7] Hung, C. C., Lillicrap, T., Abramson, J., Wu, Y., Mirza, M., Carnevale, F., ... & Wayne, G. (2019). Optimizing agent behavior over long time scales by transporting value. Nature communications, 10(1), 5223.

[8] Liu, Y., Luo, Y., Zhong, Y., Chen, X., Liu, Q., & Peng, J. (2019). Sequence modeling of temporal credit assignment for episodic reinforcement learning. arXiv preprint arXiv:1905.13420.

---

> ### Author Response · Authors · 2023-11-20
> **Additional Experiments on Decision Transformers**
>
> Based on suggestions and feedback, we have also included additional results of an online decision transformer on the Myriad environments. We include these results in Appendix F.3. Notably, decision transformers are not suitable for a purely online setting, and thus exhibits worse performance in every task than any of our action-sequence models. In fact, a simple one-step model also outperforms the online decision transformer. Please refer to the amended Appendix for additional details.

---

> ### Author Response · Authors · 2023-11-22
> **Rebuttal Response**
>
> We thank the reviewers for their initial feedback. Given that the deadline for the rebuttal period is today, we would love to hear if there is any additional answer we could provide to help the review process. Otherwise, we're hopeful that our answers addressed the main concerns from the reviewers, and hope reviewers could raise their score accordingly.
>
> Best,
>
> The Authors

---

### Author Response · Authors · 2023-11-18
**Revisions Outline**

In response to the insightful comments from the reviewers, we have made a few revisions to our paper. The revisions are highlighted in blue in the PDF for easy identification. We provide an outline of the main revisions here:

- **Related works**: We reworked the related works to better highlight the similarities and differences with prior work, pointed out by a few reviewers. A more detailed comparison is also found in Appendix G.
- **Tightness argument for Corollary 1.1**: We added an argument to ensure the tightness of Corollary 1.1, such that the bound is not meaningless. It's found in Appendix A.1
- **Additional visualization of credit assignment**: We  added a visualization of the credit assignment properties of the transformer in the toy credit assignment task in Appendix F.2.
- **Edit (20/11/2023) - Results on online Decision Transformers**: We include results of the online decision transformer as an additional baseline for the Myriad environments in Appendix F.3.

---

### Meta-Review · Area_Chair_VPK9 · 2023-12-05

**Metareview:**

The authors draw connections between policy gradients and backpropagation through deep network models. Rather than learning a one-step model, the authors propose a multi-step transition model that predicts a sequence of resulting states. Results are demonstrated on an optimal control testbed (continuous control, fully observable MDPs). The reviewers acknowledge that this is an interesting approach, with generally clear presentation of empirical and theoretical results. However, several reviewers were confused on the novelty of the paper, i.e., were the authors proposing a new perspective, or just a new model? Other reviewers cited a lack of more established empirical benchmarks, such as MuJoCo, more appropriate baselines models, and limited applicability of targeting the deterministic MDP setting. More completely addressing these concerns in the paper would strengthen it.

**Justification For Why Not Higher Score:**

While the paper presents an interesting approach, the somewhat limited applicability of the setting (relatively toy-ish problems defined by deterministic MDPs), combined with a lack of clearer baselines, leads me to conclude that the paper is not ready for publication.

**Justification For Why Not Lower Score:**

N/A

---

### Decision · Program_Chairs · 2024-01-16

Reject